# Unveiling Adversarially Robust Graph Lottery Tickets

**Subhajit Dutta Chowdhury***                                    *duttacho@usc.edu*
*Department of Electrical and Computer Engineering*
*University of Southern California, Los Angeles*

**Zhiyu Ni***                                                    *zhiyuni@usc.edu*
*Department of Electrical and Computer Engineering*
*University of Southern California, Los Angeles*

**Qingyuan Peng**                                                *pengqing@usc.edu*
*Department of Electrical and Computer Engineering*
*University of Southern California, Los Angeles*

**Souvik Kundu**                                                 *souvikku@usc.edu*
*Intel Labs, San Diego*

**Pierluigi Nuzzo**                                       *nuzzo@eecs.berkeley.edu*
*Department of Electrical Engineering and Computer Sciences, University of California, Berkeley*
*Department of Electrical and Computer Engineering, University of Southern California, Los Angeles*

**Reviewed on OpenReview:** *https://openreview.net/forum?id=PX06pUVs1P*

## Abstract

Graph lottery tickets (GLTs), comprising a sparse graph neural network (GNN) and a sparse input graph adjacency matrix, can significantly reduce the computing footprint of inference tasks compared to their dense counterparts. However, their performance against adversarial attacks remains to be fully explored. In this paper, we first investigate the resilience of GLTs against different poisoning attacks based on structure perturbations and observe that they are vulnerable and show a large drop in classification accuracy. We then present an *adversarially robust graph sparsification (ARGS)* framework that prunes the adjacency matrix and the GNN weights by minimizing a novel loss function capturing the graph homophily property and information associated with the true labels of the train nodes and the pseudo labels of the test nodes. By iteratively applying ARGS to prune both the perturbed graph adjacency matrix and the GNN model weights, we can find graph lottery tickets that are highly sparse yet achieve competitive performance under different training-time (poisoning) structure-perturbation attacks. Evaluations conducted on various benchmarks, considering attacks such as PGD, MetaAttack, PR-BCD, GR-BCD, and adaptive attack, demonstrate that ARGS can significantly improve the robustness of the generated GLTs, even when subjected to high levels of sparsity.

## 1 Introduction

Graph neural networks (GNNs) (Hamilton et al., 2017; Kipf & Welling, 2017; Veličković et al., 2018; Zhou et al., 2020; Zhang et al., 2020) achieve state-of-the-art performance on various graph-based tasks like semi-supervised node classification (Kipf & Welling, 2017; Hamilton et al., 2017; Veličković et al., 2018; Chowdhury et al., 2021), link prediction (Zhang & Chen, 2018; Chowdhury et al., 2023b), and graph classification (Ying et al., 2018). The success of GNNs is attributed to the neural message-passing scheme in which each node updates its feature by recursively aggregating and transforming the features of its neighbors. However, the

---

*Equal contributions.

effectiveness of GNNs, when scaled up to large and densely connected graphs, is impacted by the high training and inference cost and substantial memory consumption. Unified graph sparsification (UGS) (Chen et al., 2021) addresses this concern by simultaneously pruning the input graph adjacency matrix and the GNN to find a graph lottery ticket (GLT), a pair of sparse graph adjacency matrix and GNN model, which can substantially reduce inference cost without compromising model performance.

Recent studies reveal that GNNs are vulnerable to adversarial attacks (Dai et al., 2018; Wu et al., 2019; Zügner & Günnemann, 2019; Mujkanovic et al., 2022; Jin et al., 2021). In the transductive setting, the labels of a few nodes are given, and the goal is to predict the labels of the remaining nodes in the graph. An adversarial attack on the graph structure introduces unnoticeable perturbations by inserting, deleting, or rewiring edges in the graph. These perturbations increase the distribution shift between the train nodes, i.e., the nodes in the train set, and the test nodes in the test set, fooling the GNN to misclassify nodes (Li et al., 2023b) in the transductive node classification task. Many defense techniques have been developed to counter these attacks. Some techniques improve the node classification accuracy of GNNs by cleaning the perturbed graph structure (Wu et al., 2019; Entezari et al., 2020; Jin et al., 2020; Deng et al., 2022; Zhu et al., 2021b); others improve the accuracy by modifying the GNN architecture (Zhang & Zitnik, 2020; Geisler et al., 2021; Zhu et al., 2019). Although GLTs demonstrate strong performance on original benign graph data, their performance in the presence of adversarial structure perturbations remains largely unexplored. Achieving adversarially robust GLTs (ARGLTs) can enable efficient GNN inference under adversarial threats.

We pursue this objective by first investigating empirically the resilience of GLTs identified by UGS against different structure-perturbation attacks (Zügner & Günnemann, 2019; Liu et al., 2019; Mujkanovic et al., 2022; Chowdhury et al., 2024; 2023a) and showing that they are vulnerable. We then present ARGS (Adversarially Robust Graph Sparsification), an optimization framework that, given an adversarially perturbed graph, iteratively prunes the graph adjacency matrix and the GNN model weights to generate an adversarially robust graph lottery ticket (ARGLT) which achieves competitive node classification accuracy while exhibiting high levels of sparsity.

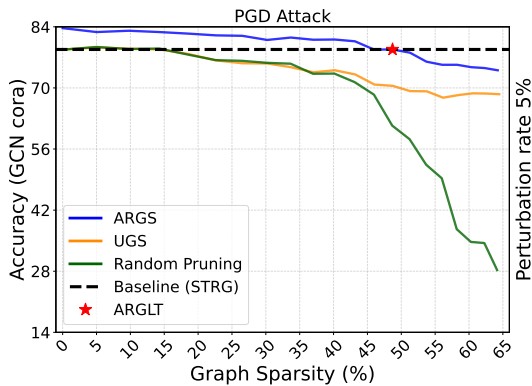

Figure 1: Comparison of different graph sparsification techniques in accuracy vs. graph sparsity. ARGS achieves similar accuracy (see red star) with 35% more sparsity for the Cora dataset under the PGD attack.

Adversarial attacks like the projected gradient descent (PGD) attack (Wu et al., 2019), the meta-learning-based graph attack (MetaAttack) (Zügner & Günnemann, 2019), the projected randomized block coordinate descent (PR-BCD) attack (Geisler et al., 2021), and the greedy randomized block coordinate descent (GR-BCD) attack (Geisler et al., 2021), poison the graph structure by adding new edges or deleting existing edges, resulting in changes in the properties of the graph. In the case of homophilic graphs, connected nodes generally have similar features and often belong to the same class, while for heterophilic graphs, linked nodes have dissimilar features and different classes. Our analysis shows that the PGD attack and the MetaAttack introduce most edge modifications around the train nodes (Li et al., 2023b) while the local structure of the test nodes is less affected. Moreover, for homophilic graphs, adversarial edges are often introduced between nodes with dissimilar features. In contrast, for heterophilic graphs, adversarial edges are introduced between nodes with dissimilar neighborhood structures. We leverage this information to formulate a new loss function that better guides the pruning of the adversarial edges in the graph and the GNN weights. Additionally, we use self-learning Amini et al. (2022) to train the pruned GNNs on sparse graph structures, which improves the classification accuracy of the GLTs. To the best of our knowledge, this is the first study on the adversarial robustness of GLTs.

Our proposal is evaluated across various GNN architectures on both homophilic (Cora, citeseer, PubMed, OGBN-ArXiv, OGBN-Products) and heterophilic (Chameleon, Squirrel) graphs attacked by the PGD attack, the MetaAttack, the PR-BCD attack (Geisler et al., 2021), and the GR-BCD attack, for the node classification task. We also evaluate the proposed technique for an adaptive attack, i.e., a stronger form of attack specifically

designed to target our technique. By iteratively applying ARGS, ARGLTs can be broadly located across the 7 graph datasets with substantially reduced inference costs, up to 98% multiply-and-accumulate (MAC) savings, and little to no accuracy drop. Figure 1 shows that, for node classification on Cora attacked by the PGD attack, our ARGLT achieves similar accuracy to that of the full models and graphs even with high graph and model sparsity of 48.68% and 94.53%, respectively. Compared to the GLTs identified by UGS, our ARGLTs on average achieve the same accuracy with 2.4× higher graph sparsity and 2.3× higher model sparsity.

## 2 Related Work

We provide an overview of the graph lottery ticket hypothesis, adversarial attacks on graphs, and the existing defenses against such attacks.

### 2.1 Graph Lottery Ticket Hypothesis

The lottery ticket hypothesis (LTH) (Frankle & Carbin, 2019) conjectures that there exist small sub-networks, dubbed as lottery tickets (LTs), within a dense randomly initialized neural network, that can be trained in isolation to achieve comparable accuracy to that of their dense counterparts. Unified graph sparsificatio (UGS) made it possible to extend the LTH to GNNs (Chen et al., 2021), showing the existence of GLTs that can make GNN inference efficient. A GNN sub-network along with a sparse input graph is defined as a GLT if the sub-network with the original initialization, trained on the sparsified graph, has a test accuracy that matches the one of the original, unpruned GNN trained on the full graph. Specifically, during training, UGS applies two differentiable binary mask tensors to the adjacency matrix and the GNN model weights, respectively. After training, the lowest-magnitude elements are removed and the corresponding mask location is updated to 0, eliminating the low-scored edges and weights from the adjacency matrix and the GNN, respectively. The sparse GNN weight parameters are then rewound to their original initialization. To identify the GLTs, the UGS algorithm is applied in an iterative fashion until pre-defined graph and weight sparsity levels are reached. Experimental results show that UGS can significantly trim down the inference computational cost without compromising predictive accuracy. In this work, we aim to find GLTs for datasets that have been adversarially perturbed. When we apply the UGS algorithm directly to the perturbed graphs, the accuracy performance of the GLTs is substantially lower than the one of their clean counterparts, calling for new methods to find adversarially robust GLTs.

### 2.2 Adversarial Attacks on Graphs

Adversarial attacks on graphs can be classified as *poisoning attacks*, perturbing the graph at train time, and *evasion attacks*, perturbing the graph at test time. Both poisoning and evasion attacks can be *targeted* or *global* attacks (Liu et al., 2019). A targeted attack deceives the model to misclassify a specific node (Zügner et al., 2018; Bojchevski & Günnemann, 2019). A global attack degrades the overall performance of the model (Zügner & Günnemann, 2019; Wu et al., 2019). Depending on the amount of information available, the existing attacks can further be categorized into *white-box* attacks, *gray-box* attacks, and *black-box* attacks (Zügner et al., 2018; Chang et al., 2020). Finally, an attacker can modify the node features, the discrete graph structure, or both. Different attacks show that structure perturbation is often more effective when compared to modifying the node features (Zhu et al., 2021a).

Examples of global poisoning attacks include the MetaAttack (Zügner & Günnemann, 2019), PGD attack (Wu et al., 2019), PR-BCD attack (Geisler et al., 2021), and GR-BCD attack (Geisler et al., 2021). Gradient-based attacks like PGD and MetaAttack treat the adjacency matrix as a parameter tensor and modify it via scaled gradient-based perturbations that aim to maximize the loss, thus resulting in degradation of the GNN prediction accuracy. PR-BCD and GR-BCD (Geisler et al., 2021) are more scalable, first-order optimization-based attacks that can scale up to large datasets like OGBN-ArXiv and OGBN-Products (Hu et al., 2020), respectively. Global poisoning attacks are highly effective in reducing the classification accuracy of multiple GNN models and are typically more challenging to counter since they modify the graph structure before training (Zhu et al., 2021a). In this work, we consider global graph-structure poisoning attacks.

### 2.3 Defenses on Graphs

Several approaches have been developed to combat adversarial attacks on graphs (Tang et al., 2020; Entezari et al., 2020; Zhu et al., 2019; Jin et al., 2020; Zhang & Zitnik, 2020; Wu et al., 2019; Deng et al., 2022; Zhou et al., 2023). Many of these techniques try to improve the classification accuracy by preprocessing the graph structure, i.e., they detect the potential adversarial edges and assign lower weights to these edges, or even remove them. Jaccard-GCN (Wu et al., 2019) removes all the edges between nodes whose features exhibit a Jaccard similarity below a certain threshold. SVD-GCN (Entezari et al., 2020) replaces the adjacency matrix with a low-rank approximation since many real-world graphs are low-rank and attacks tend to disproportionately affect the high-frequency spectrum of the adjacency matrix. ProGNN (Jin et al., 2020) leverages low-rank, sparsity, and feature smoothness properties of graphs to clean the perturbed adjacency matrix. GARNET (Deng et al., 2022) combines spectral graph embedding with probabilistic graphical models to recover the original graph topology from the perturbed graph. GNNGuard (Zhang & Zitnik, 2020) learns weights for the edges in each message passing aggregation step via cosine similarity and penalizes the adversarial edges either by filtering them out or by assigning less weight to them. STABLE (Li et al., 2022) preprocesses the graph structure by leveraging unsupervised learning. Instead of using the node features like the above mentioned techniques, STABLE leverages contrastive learning to learn new node representations, which are then used to refine the graph structure. Other techniques try to improve the GNN performance by enhancing model training through data augmentation (Li et al., 2022; Feng et al., 2020), adversarial training (Wu et al., 2019), self-learning (Li et al., 2023b), robust aggregate functions (Geisler et al., 2021; Li et al., 2022), or by developing novel GNN layers (Zhu et al., 2019).

Differently from the approaches above, GCN-LFR (Chang et al., 2021), a spectral-based method, leverages the fact that some low-frequency components in the graph spectrum are more robust to edge perturbations and regularizes the training process of a given GCN with robust information from an auxiliary regularization network to improve its adversarial performance. Overall, graph preprocessing tends to remove only a small fraction of edges from the adjacency matrix. Additionally, none of these defenses reduces the number of parameters in the GNN model, which results in unchanged computational footprints. We instead aim to improve the robustness of sparse GNNs with sparse adjacency matrices to achieve computation efficiency. As robustness generally requires more non-zero parameters, yielding parameter-efficient robust GLTs remains a challenge.

## 3 Preliminaries

**Notation.** Let $\mathcal{G} = \{\mathcal{V}, \mathcal{E}\}$ represent an undirected graph with $|\mathcal{V}|$ nodes and $|\mathcal{E}|$ edges. The topology of the graph can be represented with an adjacency matrix $\boldsymbol{A} \in \mathbb{R}^{|\mathcal{V}| \times |\mathcal{V}|}$, where $\boldsymbol{A}_{ij} = 1$ if there is an edge $e_{i,j} \in \mathcal{E}$ between nodes $v_i$ and $v_j$, while $\boldsymbol{A}_{ij} = 0$ otherwise. Each node $v_i \in \mathcal{V}$ has an attribute feature vector $\boldsymbol{x}_i \in \mathbb{R}^F$, where $F$ is the number of node features. Let $\boldsymbol{X} \in \mathbb{R}^{|\mathcal{V}| \times F}$ and $\boldsymbol{Y} \in \mathbb{R}^{|\mathcal{V}| \times C}$ denote the feature matrix and the labels of all nodes in the graph, respectively, where $C$ is the number of classes in the dataset. With a slight abuse of notation, in this paper, we will also represent a graph as a pair $\{\boldsymbol{A}, \boldsymbol{X}\}$. We call the nodes in the train set as *train nodes*, whereas those within the test set are denoted as *test nodes*. In the case of message-passing GNN, the representation of a node $v_i$ is iteratively updated by aggregating and transforming the representations of its neighbors. As an example, a two-layer GCN(Kipf & Welling, 2017) can be specified as

$$\boldsymbol{Z} = f(\{\boldsymbol{A}, \boldsymbol{X}\}, \boldsymbol{\Theta}) = \mathcal{S}(\hat{\boldsymbol{A}}\sigma(\hat{\boldsymbol{A}}\boldsymbol{X}\boldsymbol{W}_0)\boldsymbol{W}_1), \tag{1}$$

where $\boldsymbol{Z}$ is the prediction, $\boldsymbol{\Theta} = (\boldsymbol{W}_0, \boldsymbol{W}_1)$ are the weights, $\sigma(.)$ is the activation function, e.g., a rectified linear unit (ReLU), $\mathcal{S}(.)$ is the softmax function, $\hat{\boldsymbol{A}} = \tilde{\boldsymbol{D}}^{-\frac{1}{2}}(\boldsymbol{A} + \boldsymbol{I})\tilde{\boldsymbol{D}}^{-\frac{1}{2}}$ is the normalized adjacency matrix with self-loops, and $\tilde{\boldsymbol{D}}$ is the degree matrix of $\boldsymbol{A} + \boldsymbol{I}$. We consider the transductive semi-supervised node classification (SSNC) task for which the cross-entropy (CE) loss over the train nodes is given by

$$\mathcal{L}_0(f(\{\boldsymbol{A}, \boldsymbol{X}\}, \boldsymbol{\Theta})) = -\sum_{l \in \mathcal{Y}_{TL}} \sum_{j=1}^{C} \boldsymbol{Y}_{l_j} \log(\boldsymbol{Z}_{l_j}), \tag{2}$$

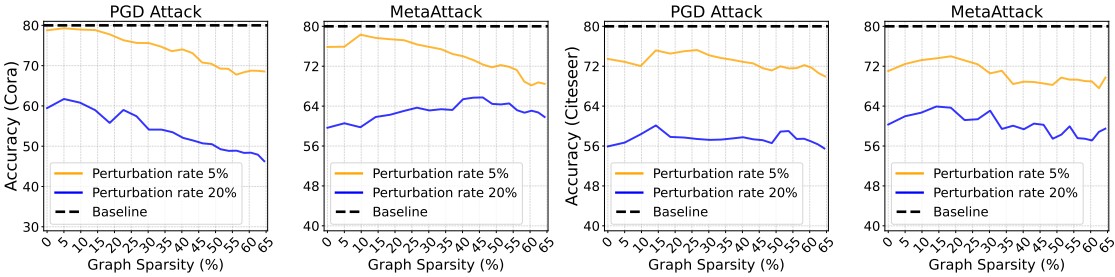

Figure 2: Classification accuracy of GLTs generated using UGS for Cora and Citeseer datasets attacked by the PGD attack and the MetaAttack. The baseline refers to accuracy on the clean graph.

where $\mathcal{Y}_{TL}$ is the set of train node indices, $C$ is the total number of classes, and $\boldsymbol{Y}_l$ is the one-hot encoded label of node $v_l$. In the transductive SSNC task, for a given graph, the labels of the train nodes are known and the goal is to predict the labels of the remaining one, i.e., the test nodes.

**Graph Lottery Tickets.** A GLT consists of a sparsified graph, obtained by pruning some edges in $\mathcal{G}$, and a GNN sub-network, with the original initialization, that can be trained to achieve comparable performance to the original GNN trained on the full graph, where performance is measured in terms of test accuracy. Given a GNN $f(\cdot, \boldsymbol{\Theta})$ and a graph $\mathcal{G} = \{\boldsymbol{A}, \boldsymbol{X}\}$, the associated GNN sub-network and the sparsified graph can be represented as $f(\cdot, \boldsymbol{m}_\theta \odot \boldsymbol{\Theta})$ and $\mathcal{G}_s = \{\boldsymbol{m}_g \odot \boldsymbol{A}, \boldsymbol{X}\}$, respectively, where $\boldsymbol{m}_g$ and $\boldsymbol{m}_\theta$ are differentiable masks applied to the adjacency matrix $\boldsymbol{A}$ and the model weights $\boldsymbol{\Theta}$, respectively, and $\odot$ is the element-wise product. UGS (Chen et al., 2021) finds the two masks $\boldsymbol{m}_g$ and $\boldsymbol{m}_\theta$ by minimizing the loss function $\mathcal{L}_{UGS} = \mathcal{L}_0(f(\{\boldsymbol{m}_g \odot \boldsymbol{A}, \boldsymbol{X}\}, \boldsymbol{m}_\theta \odot \boldsymbol{\Theta})) + \psi_1 ||\boldsymbol{m}_g||_1 + \psi_2 ||\boldsymbol{m}_\theta||_1$, such that the GNN sub-network $f(\cdot, \boldsymbol{m}_\theta \odot \boldsymbol{\Theta})$ along with the sparsified graph $\mathcal{G}_s$ can be trained to a similar accuracy as $f(, \boldsymbol{\Theta})$ on $\mathcal{G}$, where $\psi_1, \psi_2$ are the $l_1$-norm sparsity regularizers of $\boldsymbol{m}_g, \boldsymbol{m}_\theta$.

**Poisoning Attack on Graphs.** In this work, we primarily investigate the robustness of GLTs under global poisoning attacks that modify the structure of the graph. In the case of a poisoning attack, GNNs are trained on a graph that has been maliciously modified by the attacker. The aim of the attacker is to find an optimal perturbed $\boldsymbol{A'}$ that fools the GNN into making incorrect predictions. This can be formulated as a bi-level optimization problem (Zügner et al., 2018; Zügner & Günnemann, 2019):

$$\arg\max_{\boldsymbol{A'} \in \Phi(\boldsymbol{A})} \mathcal{L}_{atk}(f(\{\boldsymbol{A'}, \boldsymbol{X}\}, \boldsymbol{\Theta}^*))$$
$$\text{s.t.} \quad \boldsymbol{\Theta}^* = \arg\min_{\boldsymbol{\Theta}} \mathcal{L}_0(f(\{\boldsymbol{A'}, \boldsymbol{X}\}, \boldsymbol{\Theta})), \tag{3}$$

where $\Phi(\boldsymbol{A})$ is the set of adjacency matrices that fit the constraint $\frac{||\boldsymbol{A'} - \boldsymbol{A}||_0}{||\boldsymbol{A}||_0} \leq \Delta$, $\mathcal{L}_{atk}$ is the attack loss function, $\Delta$ is the perturbation rate, and $\boldsymbol{\Theta}^*$ is the optimal parameter of the surrogate GNN model used to perform the attack. $||\cdot||_0$ is the $L_0$ norm, counting the number of non-zero elements. We provide details about the different poisoning attacks in Appendix A.2.

## 4 Graph Sparsification Under Adversarial Attacks

We perform the MetaAttack (Zügner & Günnemann, 2019) and the PGD attack (Wu et al., 2019) on the Cora and Citeseer datasets with different perturbation rates. We use the same setup as Xu et al. (2019), Zhang & Zitnik (2020), and Mujkanovic et al. (2022) for performing the poisoning attacks on the datasets. Then, we apply UGS on these perturbed graphs to find the GLTs. As shown in Figure 2, the classification accuracy of the GLTs identified by UGS is lower than the clean graph accuracy. The difference increases substantially when the perturbation rate increases. For example, in the PGD attack, when the graph sparsity is 30%, at 5% perturbation, the accuracy drop is 6%. This drop increases to 25% when the perturbation rate is 20%. Moreover, for 20% perturbation rate, even with 0% sparsity, the accuracy of the GNN is around 20%

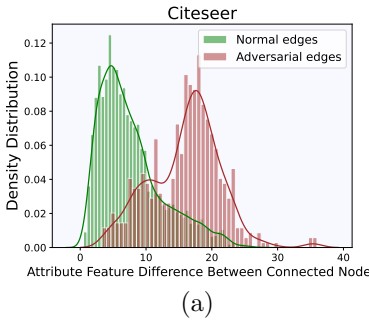 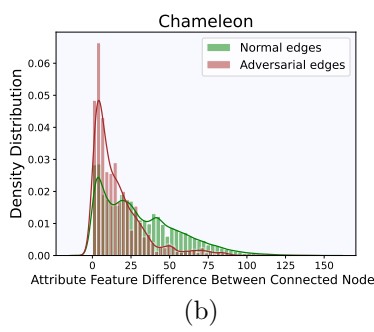 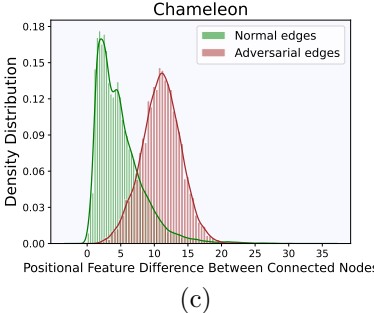

(a)                  (b)                (c)

Figure 3: Impact of adversarial attacks on graph properties. (a), (b) Density distribution of attribute feature differences of connected nodes in perturbed homophilic (Citeseer) and heterophilic (Chameleon) graphs. (c) Density distribution of positional feature differences of connected nodes in perturbed heterophilic graphs.

lower than that of the clean graph accuracy. While UGS removes edges from the perturbed adjacency matrix, as shown in Figure 2, it may not effectively remove the adversarially perturbed edges. A naïve application of UGS may not be sufficient to improve the adversarial robustness of the GLTs. Consequently, there is a need for an adversarially robust UGS technique that can efficiently remove the edges affected via adversarial perturbations while pruning the adjacency matrix and the associated GNN, along with improved adversarial training, allowing the dual benefits of improved robustness and inference efficiency.

**Analyzing the Impact of Adversarial Attacks on Graph Properties.** Adversarial attacks like the MetaAttack, PGD, PR-BCD, and GR-BCD poison the graph structure by either introducing new edges or deleting existing edges, resulting in changes in the original graph properties. We analyze the difference in the attribute features of the nodes that are connected by the clean and adversarial edges. Figure 3a and b depict the density distribution of the attribute feature difference between connected nodes in homophilic and heterophilic graph datasets attacked by the PGD attack. In homophilic graphs, the attack tends to connect nodes with large attribute feature differences. A defense technique can potentially leverage this information to differentiate between the benign and adversarial edges in the graph(Chen et al., 2022). However, this is not the case for heterophilic graphs (Zhu et al., 2022). For heterophilic graphs, we resort, instead, to the positional features of the nodes, using positional encoding techniques like DeepWalk (Perozzi et al., 2014). As we observe from Figure 3c, in heterophilic graphs, attacks tend to connect nodes with large positional feature differences. ARGS uses these graph properties to iteratively prune the adversarial edges from homophilic and heterophilic graphs.

## 5 Adversarially Robust Graph Sparsification

We present ARGS, a sparsification technique that simultaneously reduces edges in $\mathcal{G}$ and GNN parameters in $\Theta$ under adversarial attack conditions to effectively accelerate GNN inference yet maintain robust classification accuracy. ARGS reformulates the loss function to include (a) a CE loss term on the train nodes, (b) a CE loss term on a set of test nodes, and (c) a square loss term on all edges. Pruning the edges based on this combined loss function results in the removal of adversarial as well as less-important non-adversarial edges from the graph.

**Removing Edges Around the Train Nodes.** Poisoning attacks like the MetaAttack and the PGD attack tend to modify more the local structure around the train nodes than that around the test nodes (Li et al., 2023b). Specifically, a large portion of the modifications is introduced to the edges connecting a train node to a test node or a train node to another train node. We include a CE loss term associated with the train nodes, as defined in equation 2 in our objective function to account for the edges surrounding the train nodes. These edges include both adversarial and non-adversarial edges.

**Removing Adversarial Edges.** In numerous application domains, including social graphs, web page graphs, and citation graphs, connected nodes in a homophilic graph exhibit similar attribute features, while they still keep similar positional features in heterophilic graphs (Li et al., 2022; McPherson et al., 2001; Kipf & Welling, 2017). On the other hand, as shown in Figure 3, adversarial attacks tend to connect nodes with distinct attribute features in homophilic graphs and distinct positional features in heterophilic graphs. Therefore, we help remove adversarial edges and encourage feature smoothness by including the following loss to our objective function for homophilic graphs:

$$\mathcal{L}_{fs}(\boldsymbol{A}', \boldsymbol{X}) = \frac{1}{2} \sum_{i,j=1} \boldsymbol{A}'_{ij}(\boldsymbol{x_i} - \boldsymbol{x_j})^2, \tag{4}$$

where $\boldsymbol{A}'$ is the perturbed adjacency matrix and $(\boldsymbol{x_i} - \boldsymbol{x_j})^2$ measures the attribute feature difference. For heterophilic graphs, we introduce instead the following loss term:

$$\mathcal{L}_{fs}(\boldsymbol{A}') = \frac{1}{2} \sum_{i,j=1} \boldsymbol{A}'_{ij}(\boldsymbol{y_i} - \boldsymbol{y_j})^2, \tag{5}$$

where $\boldsymbol{y_i}, \boldsymbol{y_j} \in \mathbb{R}^P$ are the positional features of nodes $i, j$, obtained by running the DeepWalk algorithm (Perozzi et al., 2014) on the input graph $\mathcal{G}$, $P$ is the number of node positional features, and $(\boldsymbol{y_i} - \boldsymbol{y_j})^2$ measures the positional feature distance.

**Removing Edges Around the Test Nodes.** Removal of edges tends to be random in later iterations of UGS (Hui et al., 2023) since only a fraction of edges in $\mathcal{G}$ is related to the train nodes and directly impacts the corresponding CE loss. To better guide the edge removal around the test nodes, we also introduce a CE loss term for these nodes. However, the labels of the test nodes are unknown. We can then leverage the fact that structure poisoning attacks only modify the structure surrounding the train nodes, while their features and labels remain "clean." Therefore, we first train a simple multi-layer perceptron (MLP) with 2 layers on the train nodes. MLPs only use the node features for training. We then use the trained MLP to predict the labels for the test nodes. We call these labels *pseudo-labels*. Finally, we use the test nodes for which the MLP has high prediction confidence for computing the test node CE loss term. Let $\mathcal{Y}_{PL}$ be the set of test nodes for which the MLP prediction confidence is above a threshold and $\boldsymbol{Y}_{mlp}$ be the prediction by the MLP. The CE loss is given by

$$\mathcal{L}_1(f(\{\boldsymbol{A}', \boldsymbol{X}\}, \boldsymbol{\Theta})) = - \sum_{l \in \mathcal{Y}_{TL}} \sum_{j=1}^{C} \boldsymbol{Y}_{mlp_{l_j}} \log(\boldsymbol{Z}_{l_j}). \tag{6}$$

In summary, the complete loss function that ARGS minimizes is

$$\mathcal{L}_{ARGS} = \alpha\mathcal{L}_0(f(\{\boldsymbol{m}_g \odot \boldsymbol{A}', \boldsymbol{X}\}, \boldsymbol{m}_\theta \odot \boldsymbol{\Theta})) + \beta\mathcal{L}_{fs}(\boldsymbol{m}_g \odot \boldsymbol{A}', \boldsymbol{X})$$
$$+\gamma\mathcal{L}_1(f(\{\boldsymbol{m}_g \odot \boldsymbol{A}', \boldsymbol{X}\}, \boldsymbol{m}_\theta \odot \boldsymbol{\Theta})) + \lambda_1||\boldsymbol{m}_g||_1 + \lambda_2||\boldsymbol{m}_\theta||_1, \tag{7}$$

where $\beta, \lambda_1$, and $\lambda_2$ are hyperparameters and the value of $\alpha$ and $\gamma$ is set to 1. $\lambda_1$ and $\lambda_2$ are the $l_1$-norm regularizers of $\boldsymbol{m}_g$ and $\boldsymbol{m}_\theta$, respectively. After the training is complete, the elements with the smallest values in $\boldsymbol{m}_g$ and $\boldsymbol{m}_\theta$, representing the lowest percentages $p_g$ and $p_\theta$, are set to 0. Then, the updated masks are applied to prune $\boldsymbol{A}$ and $\boldsymbol{\Theta}$, and the weights of the GNN are rewound to their original initialization value to generate the ARGLT. We apply these steps iteratively until we reach the desired sparsity $s_g$ and $s_\theta$. Algorithm 1 illustrates our iterative pruning process. As shown in Figure 4, most of the adversarial perturbation edges are between train and test nodes (Li et al., 2023b). Moreover, our proposed sparsification technique successfully removes many of the adversarial edges. In particular, after applying our technique for 20 iterations, where each iteration removes 5% of the graph edges, the number of train-train, train-test, and test-test adversarial edges reduces by $68.13\%, 47.3\%$, and $14.3\%$, respectively.

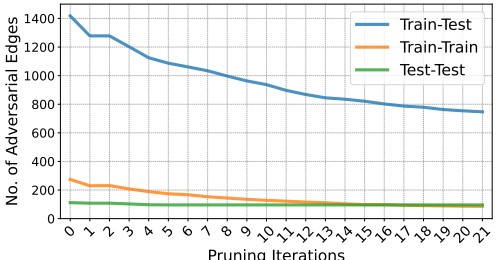

Figure 4: Evolution of adversarial edges in Cora dataset (attacked by PGD, 20% perturbation) as we apply ARGS to prune the graph. Train-Train edges connect two nodes from the train set. Train-Test edges connect a node from the train set with one from the test set. Test-Test edges connect two nodes from the test set.

---

**Algorithm 1** Adversarially Robust Graph Sparsification

---

**Input**: Graph $\mathcal{G} = \{\boldsymbol{A}, \boldsymbol{X}\}$, GNN $f(\mathcal{G}, \boldsymbol{\Theta}^0)$ with initialization $\boldsymbol{\Theta}^0$, Sparsity levels $s_g$ for graph and $s_\theta$ for GNN, Initial masks $\boldsymbol{m}_g = \boldsymbol{A}$, $\boldsymbol{m}_\theta = 1 \in \mathbb{R}^{||\boldsymbol{\Theta}^0||_0}$

**Output:** Final masks $\boldsymbol{m}_g$, $\boldsymbol{m}_\theta$

1: **while** $\left(1 - \frac{||\boldsymbol{m}_g||_0}{||\boldsymbol{A}||_0} < s_g\right)$ and $\left(1 - \frac{||\boldsymbol{m}_\theta||_0}{||\boldsymbol{\Theta}||_0} < s_\theta\right)$ **do**
2:      $\boldsymbol{m}_g^0 = \boldsymbol{m}_g$, $\boldsymbol{m}_\theta^0 = \boldsymbol{m}_\theta$, $\boldsymbol{\Theta}^0 = \{\boldsymbol{W}_0^0, \boldsymbol{W}_1^0\}$
3:      **for** $t = 0, 1, 2, \ldots, T-1$ **do**
4:          Input $\mathcal{G} = \{\boldsymbol{m}_g^t \odot \boldsymbol{A}, \boldsymbol{X}\}$ to $f(\cdot, \boldsymbol{m}_\theta^t \odot \boldsymbol{\Theta}^t)$ to compute the loss $\mathcal{L}_{ARGS}$ in equation equation 7
5:          $\boldsymbol{\Theta}^{t+1} \leftarrow \boldsymbol{\Theta}^t - \mu \nabla_{\boldsymbol{\Theta}^t} \mathcal{L}_{ARGS}$
6:          $\boldsymbol{m}_g^{t+1} \leftarrow \boldsymbol{m}_g^t - \omega_g \nabla_{\boldsymbol{m}_g^t} \mathcal{L}_{ARGS}$
7:          $\boldsymbol{m}_\theta^{t+1} \leftarrow \boldsymbol{m}_\theta^t - \omega_\theta \nabla_{\boldsymbol{m}_\theta^t} \mathcal{L}_{ARGS}$
8:      $\boldsymbol{m}_g = \boldsymbol{m}_g^{T-1}$, $\boldsymbol{m}_\theta = \boldsymbol{m}_\theta^{T-1}$
9:      Set percentage $p_g$ of the lowest-scored values in $\boldsymbol{m}_g$ to 0 and set others to 1
10:     Set percentage $p_\theta$ of the lowest-scored values in $\boldsymbol{m}_\theta$ to 0 and set others to 1

---

**Training Sparse ARGLTs.** Structure poisoning attacks do not modify the labels of the nodes. In the case of attacks like PGD and MetaAttack, the locality structure of the test nodes is less contaminated (Li et al., 2023b), implying that the train node labels and the local structure of the test nodes contain relatively clean information. We leverage this insight and train the GNN sub-network using both train nodes and test nodes. We use a CE loss term for both the train ($\mathcal{L}_0$) and test ($\mathcal{L}_1$) nodes. Since the true labels of the test nodes are not available, we train an MLP on the train nodes and then use it to predict the labels for the test nodes (Li et al., 2018; 2023b). To compute the CE loss, we use only those test nodes for which the MLP has high prediction confidence. The loss function used for training the sparse GNN on the sparse adjacency matrix generated by ARGS is

$$\min_{\boldsymbol{\Theta}} \quad \eta \mathcal{L}_0(f(\{\boldsymbol{m}_g \odot \boldsymbol{A}', \boldsymbol{X}\}, \boldsymbol{m}_\theta \odot \boldsymbol{\Theta})) + \zeta \mathcal{L}_1(f(\{\boldsymbol{m}_g \odot \boldsymbol{A}', \boldsymbol{X}\}, \boldsymbol{m}_\theta \odot \boldsymbol{\Theta})), \tag{8}$$

where $\boldsymbol{m}_\theta$ and $\boldsymbol{m}_g$ are the masks evaluated by ARGS that are kept fixed throughout training, and $\eta$ is set to 1. In the early pruning iterations, when graph sparsity is low, the test nodes are more useful in improving the model's adversarial performance because the train nodes' localities are adversarially perturbed and there exist distribution shifts between the train and test nodes. However, as the graph sparsity increases, adversarial edges associated with the train nodes are gradually removed by ARGS, thus reducing the distribution shift and making the contribution of the train nodes more important in the adversarial training.

## 6 Evaluation

We evaluate the effectiveness of ARGS and assess the existence of ARGLTs across diverse datasets and GNN models under different adversarial attacks and perturbation rates. In particular, we evaluate our sparsification method on both homophilic and heterophilic graph datasets which are attacked by two structure poisoning attacks, namely, PGD (Wu et al., 2019) and MetaAttack (Zügner & Günnemann, 2019). We consider three GNN models, namely, graph convolution networks (GCNs) (Kipf & Welling, 2017), graph isomorphism networks (GINs) (Xu et al., 2019), and graph attention networks (Veličković et al., 2018). We also evaluate ARGS on larger datasets, namely, OGBN-ArXiv and OGBN-Products (Hu et al., 2020), attacked by the PR-BCD and GR-BCD attacks, respectively, for the DeeperGCN model. Finally, we evaluate the robustness of ARGS against adaptive attacks (Mujkanovic et al., 2022).

We use DeepRobust, an adversarial attack library (Li et al., 2020), to perform the PGD attack and the MetaAttack and generate the perturbed graph adjacency matrix $\boldsymbol{A}'$. When performing these attacks, we use surrogate models which have the same type and architecture of the GNN models being attacked. For example, when attacking ARGS on a 2-layer GCN, the surrogate model is also a 2-layer GCN. We use Pytorch-Geometric (Fey & Lenssen, 2019) to perform the PR-BCD and GR-BCD attacks on the OGBN-ArXiv and OGBN-Products datasets, respectively. We compare our method with UGS (Chen et al., 2021), random pruning, and other state-of-the-art adversarial defense methods, namely, STRG (Li et al., 2023b),

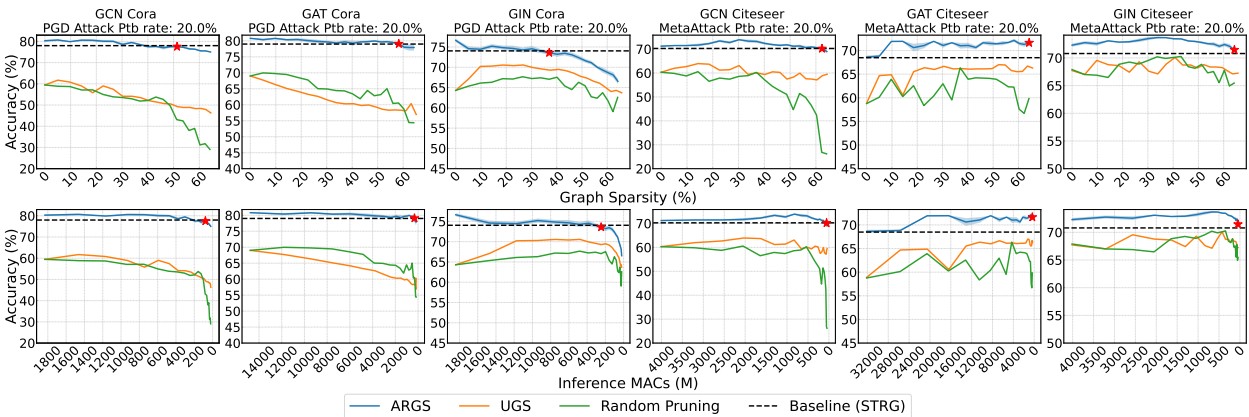

Figure 5: Node classification performance versus graph sparsity levels and inference MACs for the GCN, GIN, and GAT architectures on Cora and Citeseer datasets attacked by PGD and MetaAttack, respectively. Red stars ⋆ indicate the ARGLTs. Dash black lines represent the performance of STRG, an adversarial defense technique.

Table 1: Performance comparison between ARGS and other defense techniques in terms of accuracy and inference MAC count.

| Model | Cora (PGD attack) Perturbation Rate 20% | | Citeseer (MetaAttack) Perturbation Rate 20% | |
|---|---|---|---|---|
| | Accuracy (%) | Inference MACs (M) | Accuracy (%) | Inference MACs (M) |
| GCN-ProGNN | 63.43±0.89 | 1832.14 | 61.02 ±0.11 | 4006.91 |
| GCN-ARGS | **77.53±1.15** | **78.78** | **68.97 ±0.89** | **43.78** |
| GCN-GNNGuard | 73.19±0.72 | 1948.32 | 71.62±1.01 | 4188.33 |
| GCN-ARGS | **77.53±1.15** | **78.78** | **71.78±0.58** | **211.77** |
| GCN-GARNET | 66.66±1.10 | 1684.9 | 72.97±1.20 | 3898.21 |
| GCN-ARGS | **77.53±1.15** | **78.78** | **73.19±0.78** | **425.81** |

GARNET (Deng et al., 2022), GNNGuard (Zhang & Zitnik, 2020), ProGNN (Jin et al., 2021), and Soft Median (Geisler et al., 2021). Only UGS and random pruning techniques prune both the graph adjacency matrix and the GNN model parameters – no other existing defense techniques prune the GNN model parameters. We set $p_g = 5\%, p_\theta = 20\%$, similarly to the parameters used by UGS. More details on the dataset statistics, model configurations, and hyperparameters in ARGS can be found in Appendix A.1.

## 6.1 Defense on Homophilic Graphs

We first evaluate the performance of ARGS on homophilic graphs against PGD and MetaAttack. Due to space limitations, we show the results for a 20% perturbation rate for the Cora and Citeseer datasets. Results for PubMed and other perturbation rates are shown in Appendix A.4.

Figure 5 shows the results for the GCN, GIN, and GAT architectures on the Cora and Citeseer datasets attacked by PGD and MetaAttack, respectively, where the average accuracy of the ARGLTs is reported across 5 runs. ARGLTs at a range of graph sparsity from 30% to 60% with similar performance as the STRG baseline can be identified across the different GNN backbones. The ARGLTs significantly reduce the MAC operation count for GCN, GIN, and GAT by ∼95%, ∼97%, and ∼83%, respectively, for the Cora dataset. For the Citeseer dataset, the inference MACs reduce by ∼98% for all the backbone GNNs.

**Comparison with Other Defense Techniques.** We compare the performance of ARGS with the one of GNNGuard, GARNET, and ProGNN, which are all defense methods. Differently from ARGS, none of these methods prunes the weights of the GNN model. We compare these methods in terms of accuracy and inference MAC and we consider GCN as the backbone. For the different baselines, the GLT which has similar accuracy as the baseline with maximum graph and model sparsity is identified as the ARGLT by ARGS and reported in Table 1. For the Cora dataset, the ARGLT identified by ARGS for the PGD attack

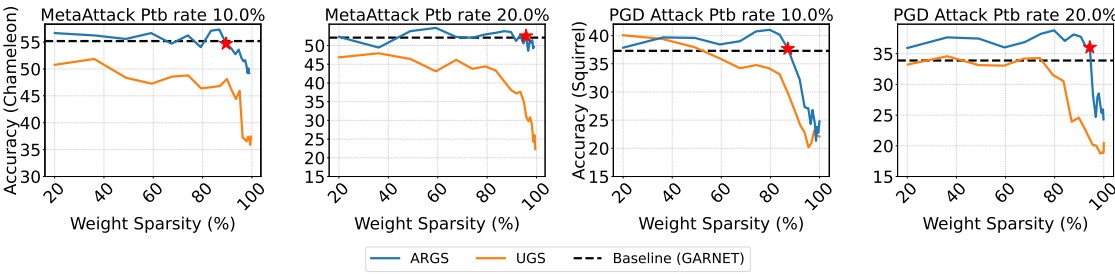

Figure 6: Node classification performance versus weight sparsity levels for GPRGNN on Chameleon and Squirrel dataset attacked by the PGD Attack and MetaAttack. ⋆ indicate the ARGLTs.

(20% perturbation rate) with maximum sparsity levels (model sparsity: 98.9%, graph sparsity: 64.1%) has a classification accuracy of 77.53%. The three different defense techniques, namely, ProGNN, GNNGuard, and GARNET have a classification accuracy of 63.43%, 73.19%, and 66.66%, respectively, which are all less than the classification accuracy of the ARGLT identified by ARGS. For the Citeseer dataset attacked by MetaAttack with a 20% perturbation rate, the most sparse ARGLT has a classification accuracy of 70.2%. The defense technique ProGNN has a classification accuracy of 61.02%, which is less than the classification accuracy of the most sparse ARGLT. The defense techniques GNNGuard and GARNET have a classification accuracy of 71.62% and 72.97%, respectively. In these cases, the GLT with the same classification accuracy as the defense technique is reported in Table 1.

## 6.2 Defense on Heterophilic Graphs

We report the classification accuracy of ARGS on heterophilic graphs in Figure 6. We use GPRGNN (Chien et al., 2020) as the GNN model for the heterophilic graph datasets Chameleon and Squirrel (McCallum et al., 2000). GPRGNN performs better than GCN, GIN, and GAT for heterophilic graphs (Deng et al., 2022). We use GARNET as the baseline since it achieves state-of-the-art adversarial classification accuracy compared to other defense techniques for heterophilic graphs. As shown in Figure 6, ARGS is able to identify GLTs that achieve similar classification accuracy as GARNET for the Chameleon and Squirrel datasets attacked by PGD and MetaAttack with 85% to 97% weight sparsity, resulting in a substantial reduction in inference MACs.

## 6.3 Defense on Larger Graphs

We evaluate the robustness of ARGS on the large-scale datasets OGBN-ArXiv and OGBN-Products. OGBN-ArXiv has 170,000 nodes and 1.16 million edges while OGBN-Products has 2.5 million nodes and 61 million edges. We use the PR-BCD attack for perturbing the OGBN-ArXiv dataset. Attempting the PR-BCD attack on the OGBN-Products dataset resulted in out-of-memory errors. We then conducted a more scalable GR-BCD attack (Geisler et al., 2021) on the OGBN-Products dataset, employing a perturbation rate of 50%. The reference GNN model is Deeper-GCN (Li et al., 2023a). For both the PR-BCD and GR-BCD attacks, the adversarial edges are uniformly distributed among the train and test nodes. Hence, we set the value of $\zeta$ to be 0. The PGD attack or MetaAttack faces timeout due to memory for these large graphs. Figure 7 shows that ARGS with 28-layer DeeperGCN can identify ARGLTs that have higher model and graph sparsity compared to UGS. We take GARNET as the baseline since it achieves better adversarial robustness than other defense techniques for the OGBN-ArXiv dataset. GARNET uses 28-layer DeeperGCN as the backbone

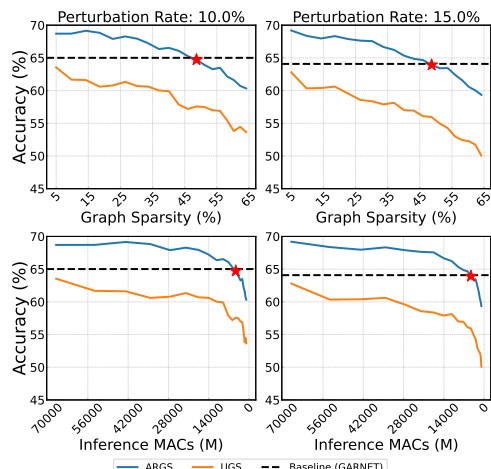

Figure 7: Node classification performance versus graph sparsity levels and inference MACs for Deeper-GCN on OGBN-ArXiv dataset attacked by PR-BCD. ⋆ indicate the ARGLTs.

Table 2: Comparison between ARGLTs identified by ARGS and GLTs identified by UGS in terms of Graph sparsity, Model sparsity, and Inference MACs across different baselines for the OGBN-Products dataset

| Baseline | Accuracy (%) | Graph Sparsity(%) | | Model Sparsity(%) | | Inference MACs(M) | |
|---|---|---|---|---|---|---|---|
| | | UGS | ARGS | UGS | ARGS | UGS | ARGS |
| Soft Median GDC | 59.23 | 87.84 | 87.84 | 87.89 | 87.89 | 75721.5 | 75727.8 |
| GCN | 62.71 | 83.32 | **87.84** | 83.36 | **87.89** | 104076.4 | **75727.8** |
| GNNGuard | 63.22 | 83.32 | **87.84** | 83.36 | **87.89** | 104076.4 | **75727.8** |
| GARNET | 74.97 | 19.00 | **40.95** | 19.01 | **40.98** | 506511.0 | **369089.2** |

Table 3: ARGS and UGS performance comparison for the PGD attack and an adaptive attack for the Cora, Citeseer, and PubMed dataset. GCN is used as the GNN model.

| Dataset | Technique | Attack | Classification Accuracy at Perturbation Rate 5% | | Classification Accuracy at Perturbation Rate 10% | |
|---|---|---|---|---|---|---|
| | | | Graph Sparsity 22.64% Model Sparsity 67.60% | Graph Sparsity 43.16% Model Sparsity 91.70% | Graph Sparsity 22.64% Model Sparsity 67.60% | Graph Sparsity 43.16% Model Sparsity 91.70% |
| Cora | ARGS | PGD Attack | **82.04±1.09** | **80.68±0.85** | **82.8±0.77** | **80.18±1.13** |
| | | Adaptive Attack | 80.33±1.35 | 78.77±1.86 | 79.68±1.35 | 77.16±0.98 |
| Citeseer | ARGS | PGD Attack | **75.32±0.88** | **74.17±0.56** | **74.7±0.98** | **73.53±1.05** |
| | | Adaptive Attack | 74.11±1.76 | 73.16±0.87 | 72.93±1.87 | 71.89±1.01 |
| PubMed | ARGS | PGD Attack | **85.57±0.07** | **85.41±0.09** | **82.72±0.05** | **83.36±0.10** |
| | | Adaptive Attack | 83.09±0.06 | 83.75±0.09 | 80.78±0.12 | 83.04±0.09 |

GNN. The model sparsity and graph sparsity of the ARGLT are 94.50% and 48.67%, respectively, for the 10% perturbed dataset, and 94.50% and 48.70%, respectively, for the 15% perturbed dataset. Results on the OGBN-Products dataset are reported in Table 2. Our baselines include GCN, GARNET, GNNGuard, and Soft Median GDC. For the different baselines, the GLT which has similar accuracy as the baseline with maximum graph and model sparsity is identified as the ARGLT by ARGS and reported in Table 2. As evident from Table 2, ARGS can identify lottery tickets with higher graph and model sparsity than UGS.

## 6.4  Defense Against Adaptive Attacks

Recently, adaptive attacks (Mujkanovic et al., 2022) have been developed, which are stronger attacks since they are specifically tailored for a given defense technique. Because all the components in the loss function of ARGS are differentiable, ARGS can be directly attacked by an adaptive attack. Specifically, we evaluate ARGS on a gradient-based adaptive attack, called Meta-PGD (Mujkanovic et al., 2022) which iteratively perturbs the adjacency matrix. Table 3 compares the performance of ARGS against the PGD attack and the adaptive attack, with GCN as the GNN backbone for Cora and CiteSeer. For a 5% perturbation rate, the accuracy of the ARGLT identified by ARGS reduces by only ∼1.7% for Cora while for CiteSeer it reduces by only ∼1.5%. For a 10% perturbation rate, the reduction in classification accuracy is ∼2.9% for the Cora dataset and ∼2.5% for the Citeseer dataset, showing that the performance degradation of ARGS from adaptive attacks is minimal. We also perform the adaptive attack on OGBN-ArXiv and the results are highlighted in Table 4.

Table 4: ARGS and UGS performance comparison for the PRBCD attack and adaptive attack for the OGBN-ArXiv dataset. DeeperGCN is used as the GNN model.

| Dataset | Technique | Attack | Classification Accuracy at Perturbation Rate 10% | | Classification Accuracy at Perturbation Rate 15% | |
|---|---|---|---|---|---|---|
| | | | Graph Sparsity 5.0% Model Sparsity 20.0% | Graph Sparsity 26.5% Model Sparsity 73.8% | Graph Sparsity 5.0% Model Sparsity 20.0% | Graph Sparsity 26.5% Model Sparsity 73.8% |
| OGBN-ArXiv | ARGS | PRBCD Attack | **68.70** | **68.28** | **69.19** | **67.64** |
| | | Adaptive Attack | 68.30 | 67.83 | 67.06 | 66.38 |

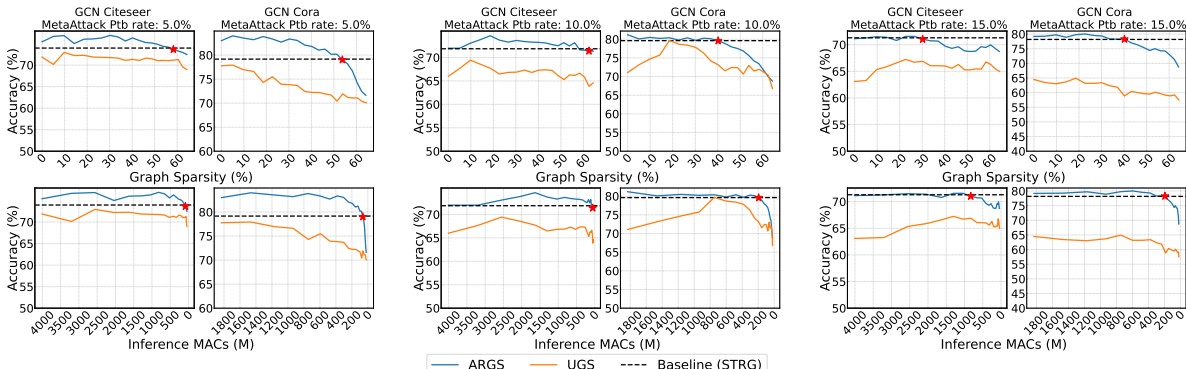

Figure 8: Node classification performance over achieved graph sparsity levels for a GCN model on Cora and Citeseer datasets attacked by a version of the MetaAttack that modifies both graph structure and node features. The perturbation rates are 5%, 10%, and 15%. Red stars ⋆ indicate the ARGLTs. STRG is used as the baseline. ⋆ indicate the ARGLTs.

## 6.5 Analysis Under Structure and Node Feature Attacks

In addition to structural attacks, we also evaluate the performance of ARGS against an attack that modifies both the graph structure and the node features simultaneously. Mettack (Zügner & Günnemann, 2019) can be modified for this purpose. For a given perturbation budget, this attack performs a structure perturbation and a feature perturbation at each iteration, and between the two perturbations, it chooses the one that results in a higher attack loss. This iterative process is repeated until the perturbation budget is exhausted. We attack the Cora and Citeseer datasets with 5%, 10%, and 15% perturbation rates, and use the STRG defense technique as the baseline. ARGS can find highly sparse GLTs that achieve similar classification accuracy as the baseline for different graph datasets perturbed with different perturbation rates using this attack. For example, for a 5% perturbation rate, ARGS finds GLTs that have 53.75% graph sparsity and 96.55% model sparsity for the Cora dataset and 58.31% graph sparsity and 97.92% model sparsity for the Citeseer dataset. We also include the performance of UGS for comparison. Figure 8 shows that, for the same sparsity levels, GLTs identified by ARGS achieve much higher classification accuracy when compared to GLTs identified by UGS. We observe that the attacked graph contains more edge perturbations than feature perturbations since modifying the graph structure results in higher attack loss than modifying the node features. This result shows that ARGS can find highly sparse GLTs for graphs attacked by both structure and node feature perturbations.

## 6.6 Ablation Study

We evaluate the effectiveness of each component of the proposed loss function for the sparsification algorithm by performing an ablation study, as shown in Table 5. We consider the Cora dataset under the PGD attack with 10% and 20% perturbation rates. Configuration 1 corresponds to ARGS with all the loss components in (7). Configuration 2 does not use the feature smoothness component in (4) while configuration 3 skips the CE loss associated with the predicted test nodes in (6). Configuration 4 skips both the smoothness and CE loss on predicted test nodes. Table 5 shows that both configurations 2 and 3 improve the final performance when compared to that of configuration 4, highlighting the importance of the losses introduced in (4) and (6). More importantly, at both high and low target sparsity, we yield the best classification performance with configuration 1, showcasing the importance of the unified loss function in (7). Further ablation studies are provided in Appendix A.6.

Table 5: Ablation study.

| GCN, Cora, PGD Attack | | | | | | Classification Accuracy at Perturbation Rate 10% | | Classification Accuracy at Perturbation Rate 20% | |
|---|---|---|---|---|---|---|---|---|---|
| Configuration | $\alpha$ | $\beta$ | $\gamma$ | $\eta$ | $\zeta$ | Graph Sparsity 9.8% Model Sparsity 36.1% | Graph Sparsity 64.4% Model Sparsity 98.9% | Graph Sparsity 9.8% Model Sparsity 36.1% | Graph Sparsity 64.5% Model Sparsity 98.9% |
| 1 | ✓ | ✓ | ✓ | ✓ | ✓ | **83.25** | **75.10** | **80.63** | **75.60** |
| 2 | ✓ | ✗ | ✓ | ✓ | ✓ | 82.04 | 70.57 | 78.92 | 64.84 |
| 3 | ✓ | ✓ | ✗ | ✓ | ✓ | 82.44 | 72.84 | 78.97 | 52.92 |
| 4 | ✓ | ✗ | ✗ | ✓ | ✓ | 80.58 | 62.42 | 75.7 | 54.18 |

# 7 Conclusion

In this paper, we first empirically observed that the performance of GLTs collapses against structure-perturbation poisoning attacks. To address this issue, we presented a new adversarially robust graph sparsification technique, ARGS, that prunes the perturbed adjacency matrix and the GNN weights by minimizing a novel loss function. By iteratively applying ARGS, we found ARGLTs that are highly sparse yet achieve competitive performance under different structure poisoning attacks. Our evaluation showed the superiority of our method over UGS at both high and low sparsity regimes.

## 7.1 Broader Impact

Graphs are universally used to capture the structure of real-world complex systems. Empowering deep learning for reasoning and making predictions over graph-structured data is of broad interest in a wide range of applications, such as recommendation systems, neural architecture search, and drug discovery. However, scaling up GNNs to large datasets is often difficult due to the computational costs. Besides, GNNs are highly vulnerable to adversarial attacks. This work aims to find GNN models that reduce the computational footprint yet maintain adversarial robustness. Such robust models will enable efficient and reliable deployment of GNNs.

## 7.2 Acknowledgments

This research was supported in part by the National Science Foundation under Awards 1846524 and 2139982, the Office of Naval Research under Award N00014-20-1-2258, the Okawa Research Grant, and the USC Center for Autonomy and Artificial Intelligence.

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

# A Appendix

In this section, we provide details about the datasets, different attacks considered, and additional results.

## A.1 Dataset Details

We use seven benchmark datasets, namely, Cora, Citeseer, PubMed, OGBN-arXiv, OGBN-Produts, Chameleon, and Squirrel, to evaluate the efficacy of ARGS. Details about the datasets are summarized in Table 6. In the case of Cora, Citeseer, and PubMed, 10% of the data constitutes the train set, 10% of the data constitutes the validation set, while the test set is the remaining 80%. For Chameleon and Squirrel, we keep the same data split settings as Chien et al. (2020). For OGBN-ArXiv and OGBN-Products we follow the data split setting of the Open Graph Benchmark (OGB) (Hu et al., 2020).

## A.2 Poisoning Attacks on GNNs

This section gives further details about the poisoning attacks introduced in the paper. We consider different structure poisoning attacks to perturb the input graph. We recall the attacker's objective is to find an optimal perturbed $A'$ which results in degradation in the performance of the GNN model on the test data. Poisoning attacks can be formulated as bi-level optimization problems, as shown in (3). The MetaAttack tackles the

Table 6: Details on the datasets.

| Datasets | Type | #Nodes | #Edges | Classes | Features |
|---|---|---|---|---|---|
| Cora | Homophilic | 2485 | 5069 | 7 | 1433 |
| Citeseer | Homophilic | 2110 | 3668 | 6 | 3703 |
| PubMed | Homophilic | 19717 | 44338 | 3 | 500 |
| OGBN-ArXiv | Homophilic | 169,343 | 1,166,243 | 40 | 128 |
| OGBN-Products | Homophilic | 2,449,029 | 61,859,140 | 47 | 100 |
| Chameleon | Heterophilic | 2277 | 62792 | 5 | 2325 |
| Squirrel | Heterophilic | 5201 | 396846 | 5 | 2089 |

bi-level problem using meta-gradients (Bojchevski & Günnemann, 2019). It treats the graph adjacency matrix as a hyperparameter which is to be optimized such that $\mathcal{L}_{atk}$ increases. The PGD attack (Xu et al., 2019) relaxes the discrete adjacency matrix $\boldsymbol{A}$ to a continuous matrix in $\{0,1\}^{n \times n}$ during the gradient-based optimization and optimizes its entries. The resulting adjustments to the matrix entries reflect the probability of an edge getting flipped. After each gradient update, the entries in the adjacency matrix are projected back such that the perturbations are within $\Delta$. Out of the different perturbed adjacency matrices in $\Phi(\boldsymbol{A})$, the one which results in maximum $\mathcal{L}_{atk}$ is chosen as the perturbed graph $\boldsymbol{A'}$. Attacks like PGD and MetaAttack face scalability issues when extended to large graph datasets like OGBN-ArXiv and OGBN-Products. Inspired by the randomized block coordinate descent (RBCD), more scalable attacks have recently been developed, namely, the PR-BCD and GR-BCD attacks Geisler et al. (2021), where only a subset of variables is optimized at a time, and only the gradients of those variables are computed, resulting in lower memory requirements.

## A.3 ARGS Implementation Details

We follow the setup used by UGS as our default setting (Chen et al., 2021) for ARGS. For Cora, Citeseer, and PubMed, we conduct all our experiments on two-layer GCN, GIN, and GAT networks with 512 hidden units. The graph sparsity $p_g$ and model sparsity $p_\theta$ are 5% and 20% unless otherwise stated. The value of $\beta$ is chosen from $\{0.01, 0.1, 1, 10\}$ while the value of $\alpha, \gamma, \eta$, and $\zeta$ is 1 by default. We use the Adam optimizer for training the GNNs. In each pruning round, the number of epochs to update the masks is by default 200, using early stopping. The 2-layer MLP used for predicting the pseudo-labels of the test nodes has by default hidden dimension of 1024 unless otherwise mentioned. We use DeepRobust, an adversarial attack library (Li et al., 2020), to implement the PGD attack and the MetaAttack on the different datasets. We use Pytorch-Geometric (Fey & Lenssen, 2019) to perform the PR-BCD and GR-BCD attacks on the OGBN-ArXiv and OGBN-Products datasets, respectively. All the experiments are conducted on an NVIDIA Tesla A100 (80-GB GPU).

For GCN, the values of $\lambda_1$ and $\lambda_2$ are both $10^{-2}$ for Cora and Citeseer, while for PubMed they are $10^{-6}$ and $10^{-3}$, respectively. The value of the learning rate is $8 \times 10^{-3}$ and that of the weight decay is $8 \times 10^{-5}$ for the Cora dataset. For Citeseer and PubMed, the learning rate is $10^{-2}$ and the weight decay is $5 \times 10^{-4}$. For the different datasets, we use a dropout of 0.5.

In the case of GIN, for the Cora dataset, the learning rate is $8 \times 10^{-3}$, the weight decay is $8 \times 10^{-5}$, $\lambda_1$ is $10^{-3}$, and $\lambda_2$ is $10^{-3}$. For Citeseer, the learning rate is $10^{-2}$, the weight decay is $5 \times 10^{-4}$, $\lambda_1$ is $10^{-5}$, and $\lambda_2$ is $10^{-5}$. For GAT, in the case of the Cora dataset, the learning rate is $8 \times 10^{-3}$, the weight decay is $8 \times 10^{-5}$, $\lambda_1$ is $10^{-3}$, $\lambda_2$ is $10^{-3}$, and dropout is 0.6. Finally, for the Citeseer dataset, the learning rate is $10^{-2}$, the weight decay is $5 \times 10^{-4}$, $\lambda_1$ is $10^{-5}$, $\lambda_2$ is $10^{-5}$, and the dropout is 0.6.

We use DeeperGCN models for the OGBN-ArXiv and OGBN-Products datasets. In the case of OGBN-ArXiv, we use a 28-layer DeeperGCN model; for OGBN-Products, we use a 14-layer DeeperGCN model. For OGBN-ArXiv, the learning rate is $10^{-2}$, $\lambda_1$ is $10^{-6}$, and $\lambda_2$ is $10^{-6}$. For OGBN-Products, the learning rate is $10^{-3}$, $\lambda_1$ is $10^{-4}$, and $\lambda_2$ is $10^{-6}$. Droput is 0.5 for both the datasets. In the case of heterophilic graphs, we use GPRGNN as the backbone GNN. For the Chameleon dataset, the learning rate is $5 \times 10^{-2}$, $\lambda_1$ is $10^{-3}$, $\lambda_2$ is $10^{-3}$, and dropout is 0.4. For the Squirrel dataset, the learning rate is $5 \times 10^{-2}$, $\lambda_1$ is $10^{-6}$, $\lambda_2$ is $10^{-2}$, and dropout is 0.4.

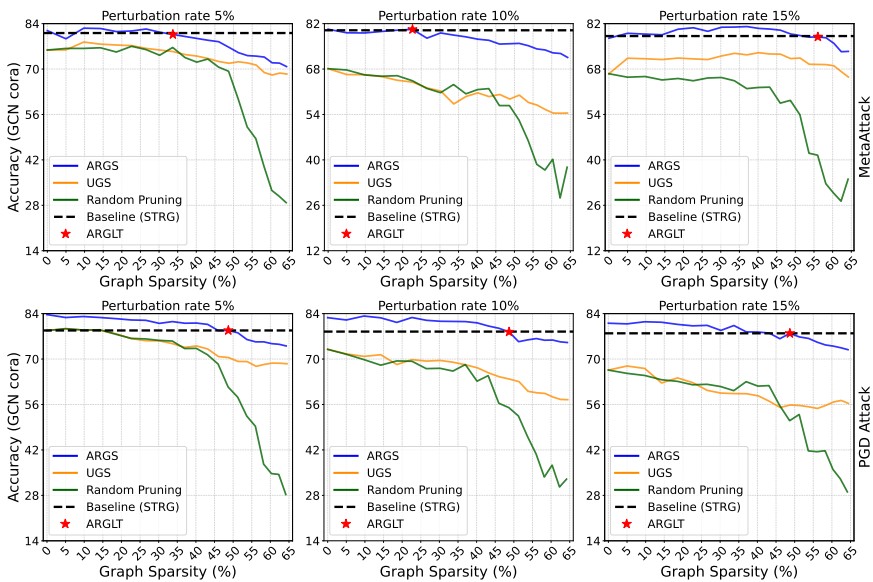

Figure 9: Node classification performance over achieved graph sparsity levels for GCN on the Cora dataset attacked by the PGD attack and MetaAttack. The perturbation rates are 5%, 10%, and 15%. Red stars ⋆ indicate the ARGLTs which achieve similar performance with high sparsity. STRG is used as the baseline.

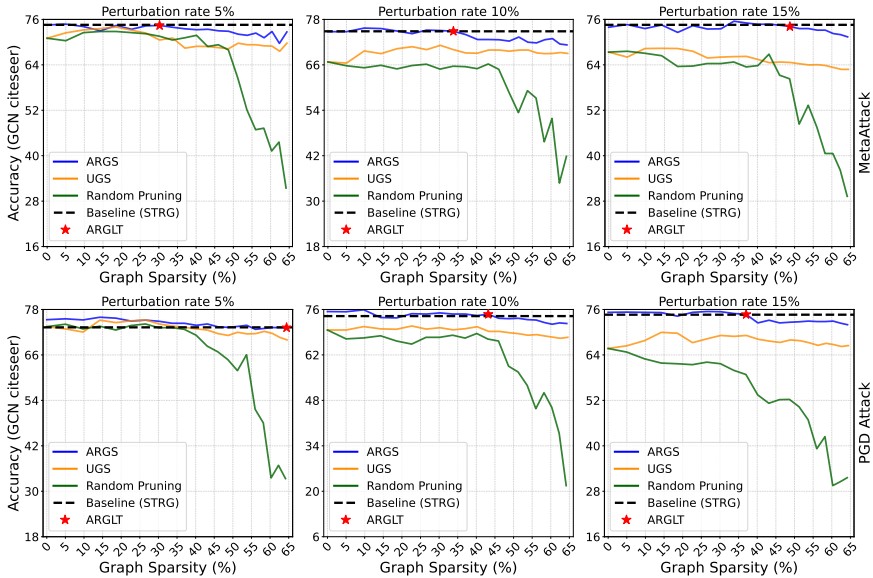

Figure 10: Node classification performance over achieved graph sparsity levels for GCN on the Citeseer dataset attacked by the PGD attack and the MetaAttack. The perturbation rates are 5%, 10%, and 15%. Red stars ⋆ indicate the ARGLTs which achieve similar performance as that of the baseline with high sparsity. STRG is used as the baseline.

## A.4 Performance Evaluation of ARGS: Additional Results

We evaluate the performance of ARGS on homophilic graphs perturbed by the PGD attack and MetaAttack. We show the results for a 5%, 10%, and 15% perturbation rate for the Cora and Citeseer datasets in Figures 9 and 10. Figures 11 and 12 show the performance of ARGS on the Cora and Citeseer datasets with GIN as the backbone GNN. Finally, Figure 13 shows the performance of ARGS on Citeseer with GAT as the backbone GNN. Finally, we evaluate the performance of ARGS on the PubMed dataset, as shown in Figure 14.

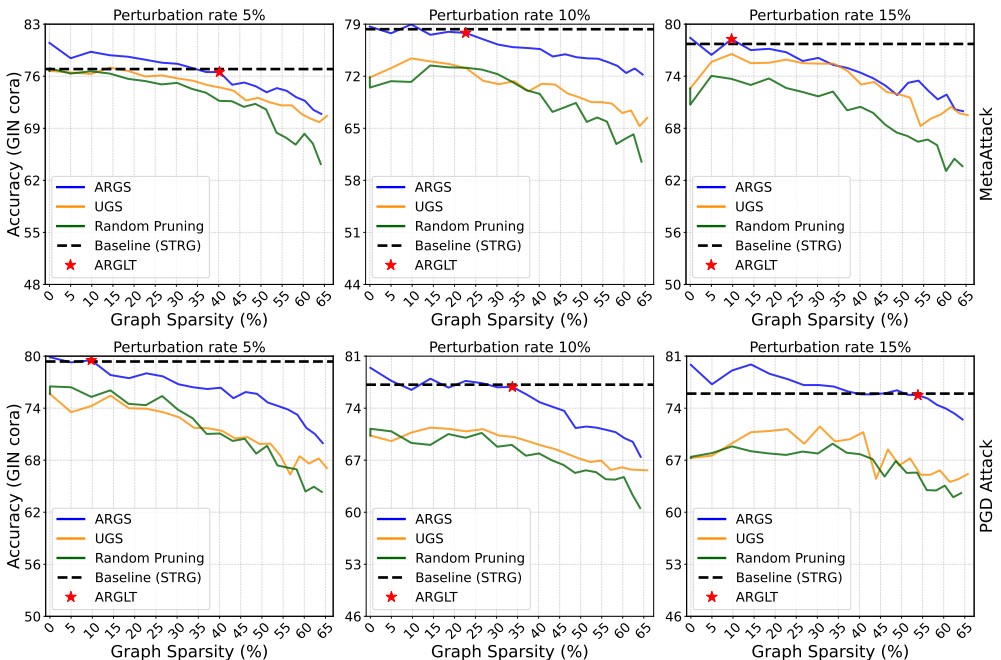

Figure 11: Node classification performance over achieved graph sparsity levels for GIN on the Cora dataset attacked by the PGD attack and the MetaAttack. The perturbation rates are 5%, 10%, and 15%. Red stars ⋆ indicate the ARGLTs which achieve similar performance with high sparsity. STRG is used as the baseline.

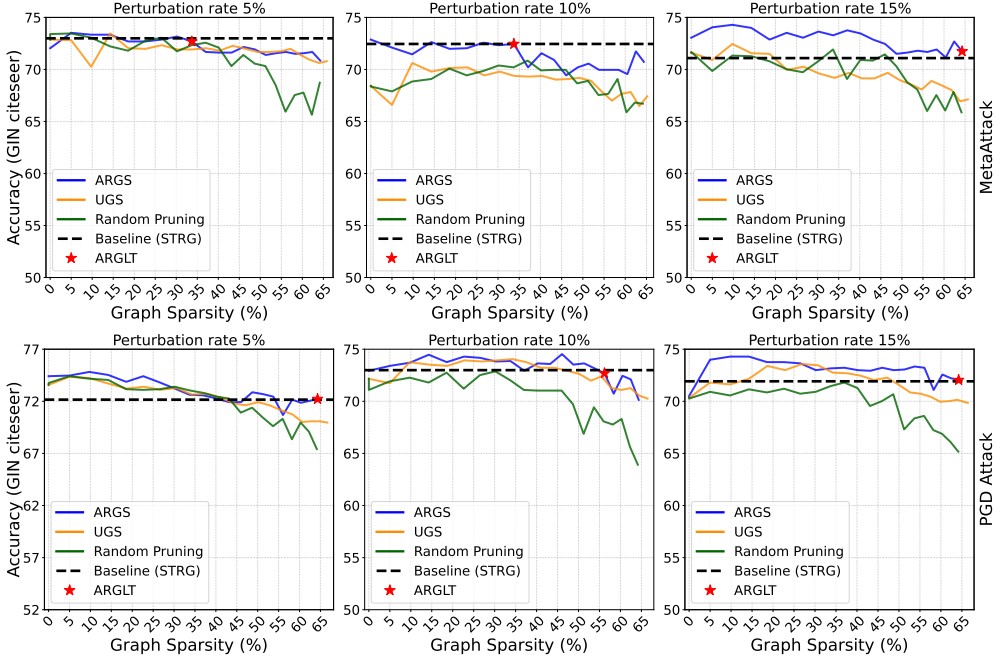

Figure 12: Node classification performance over achieved graph sparsity levels for GIN on the Citeseer dataset attacked by the PGD attack and the MetaAttack. The perturbation rates are 5%, 10%, and 15%. Red stars ⋆ indicate the ARGLTs which achieve similar performance with high sparsity. STRG is used as the baseline.

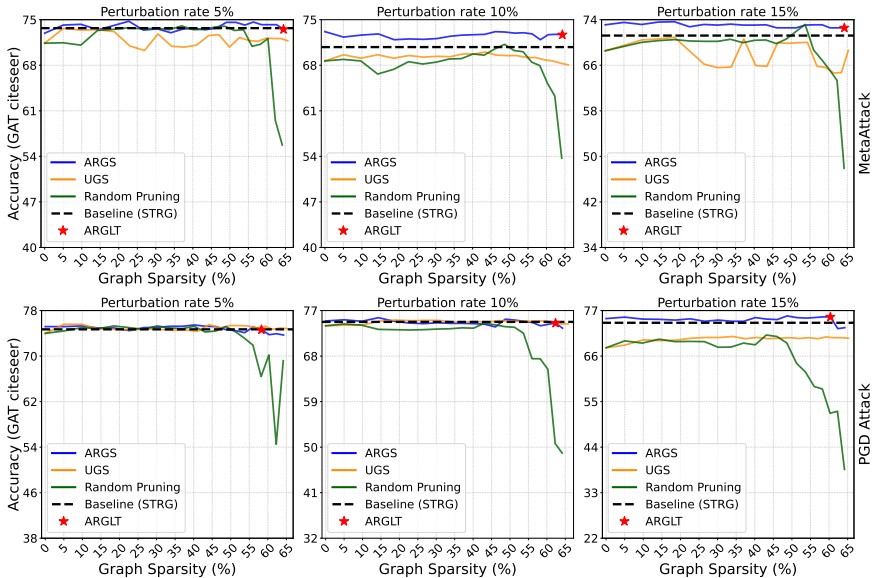

Figure 13: Node classification performance over achieved graph sparsity levels for GAT on the Citeseer dataset attacked by the PGD attack and the MetaAttack. The perturbation rates are 5%, 10%, and 15%. Red stars ⋆ indicate the ARGLTs which achieve similar performance with high sparsity. STRG is used as the baseline.

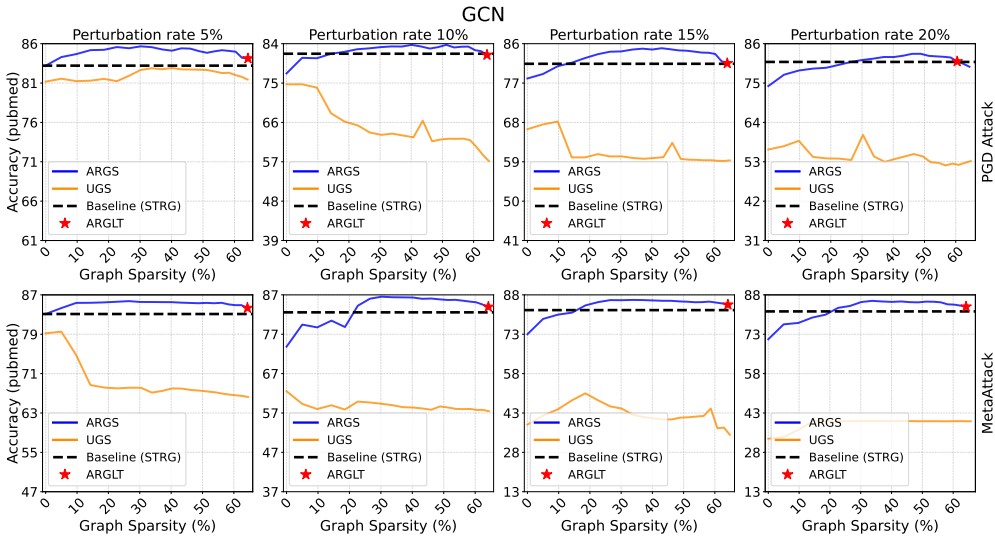

Figure 14: Node classification performance over achieved graph sparsity levels for GCN on the PubMed dataset attacked by the PGD attack and the MetaAttack. The perturbation rates are 5%, 10%, 15%, and 20%. Red stars ⋆ indicate the ARGLTs which achieve similar performance with high sparsity. STRG is used as the baseline.

## A.5 Comparing the Impact of Node Attribute Features and Positional Features on ARGLTs for Homophilic Graphs

For homophilic graphs, we considered node attribute features for removing the adversarial edges, while node positional features were considered for heterophilic graphs. We then perform a set of experiments, where we instead consider positional features of the nodes in homophilic graphs. We observe that the overlap between the density distribution of positional feature differences for clean edges and that for adversarial edges is higher when compared to the overlap between the density distributions of attribute features. However, the two density distributions of positional feature differences are still separable. The results are reported in Table 7.

Table 7: ARGS performance comparison when positional and attribute features are considered in the loss function for homophilic graphs in the Cora and Citeseer datasets on three different GNN models.

| | | | | GCN | | GIN | | GAT | |
|---|---|---|---|---|---|---|---|---|---|
| Dataset | Attack | Perturbation rate | Feature Type | Graph Sparsity | Model Sparsity | Graph Sparsity | Model Sparsity | Graph Sparsity | Model Sparsity |
| Cora | PGD | 5% | Attribute | **49.2%** | **94.8%** | **58.3%** | **97.2%** | **64.2%** | **98.9%** |
| | | | Positional | 46.0% | 93.3% | **58.3%** | **97.2%** | **64.2%** | **98.9%** |
| | | 10% | Attribute | **48.3%** | **94.2%** | **46.0%** | **93.5%** | **64.3%** | **98.9%** |
| | | | Positional | 37.2% | 87.1% | 40.1% | 89.8% | **64.3%** | **98.9%** |
| | | 15% | Attribute | **49.1%** | **94.8%** | **53.7%** | **96.7%** | **64.2%** | **98.9%** |
| | | | Positional | 37.2% | 87.1% | 18.5% | 59.2% | **64.3%** | **98.9%** |
| Citeseer | Mettack | 5% | Attribute | **31.2%** | **79.4%** | **33.6%** | **83.6%** | **64.2%** | **98.9%** |
| | | | Positional | 25.8% | 74.2% | 26.5% | 74.2% | **64.2%** | **98.9%** |
| | | 10% | Attribute | **33.6%** | **83.6%** | **64.2%** | **98.8%** | **64.2%** | **98.8%** |
| | | | Positional | **33.6%** | **83.6%** | **64.2%** | **98.8%** | **64.2%** | **98.8%** |
| | | 15% | Attribute | **48.7%** | **94.8%** | **64.2%** | **98.8%** | **64.3%** | **98.8%** |
| | | | Positional | 43.1% | 91.9% | **64.2%** | **98.8%** | **64.3%** | **98.8%** |

When positional features are considered, ARGS is still able to find highly sparse GLTs, but the sparsity levels of the GLTs in some cases are lower than those obtained when attribute features are considered.

## A.6 Further Ablation Study

We perform an ablation study to verify the effectiveness of each component of the proposed loss function used for the sparsification algorithm. Part of this study is already included in the main sections of the paper 5. We report the rest of the results in Table 8. In particular, we present the analysis performed on the Cora dataset for all the 3 different attacks with all the 4 perturbation rates. We recall that configuration 1 corresponds to ARGS with all the loss components. As shown in Table 8, at both high and low target sparsity, we yield the best classification performance with configuration 1, showcasing the importance of all the components of the proposed loss function.

Table 8: Ablation study.

| GCN, Cora, PGD Attack | | | | | | Classification Accuracy at Perturbation Rate 5% | | Classification Accuracy at Perturbation Rate 15% | |
|---|---|---|---|---|---|---|---|---|---|
| Configuration | $\alpha$ | $\beta$ | $\gamma$ | $\eta$ | $\zeta$ | Graph Sparsity 22.7% Model Sparsity 67.7% | Graph Sparsity 60.4% Model Sparsity 98.2% | Graph Sparsity 22.7% Model Sparsity 67.7% | Graph Sparsity 60.4% Model Sparsity 98.2% |
| 1 | ✓ | ✓ | ✓ | ✓ | ✓ | **82.04** | **74.75** | **80.23** | **73.99** |
| 2 | ✓ | ✗ | ✓ | ✓ | ✓ | 81.84 | 73.64 | 79.98 | 68.81 |
| 3 | ✓ | ✓ | ✗ | ✓ | ✓ | 81.69 | 74.45 | 76.86 | 72.89 |
| 4 | ✓ | ✗ | ✗ | ✓ | ✓ | 79.28 | 71.33 | 74.70 | 63.48 |
| GCN, Cora, Mettack Attack | | | | | | Classification Accuracy at Perturbation Rate 5% | | Classification Accuracy at Perturbation Rate 10% | |
| Configuration | $\alpha$ | $\beta$ | $\gamma$ | $\eta$ | $\zeta$ | Graph Sparsity 22.7% Model Sparsity 67.6% | Graph Sparsity 62.3% Model Sparsity 98.6% | Graph Sparsity 22.6% Model Sparsity 67.5% | Graph Sparsity 64.2% Model Sparsity 98.9% |
| 1 | ✓ | ✓ | ✓ | ✓ | ✓ | **81.74** | **71.83** | **80.23** | **71.58** |
| 2 | ✓ | ✗ | ✓ | ✓ | ✓ | 80.89 | 69.91 | 78.17 | 70.98 |
| 3 | ✓ | ✓ | ✗ | ✓ | ✓ | 79.88 | 71.33 | 75.40 | 66.81 |
| 4 | ✓ | ✗ | ✗ | ✓ | ✓ | 78.89 | 69.03 | 75.40 | 60.97 |
| GCN, Cora, Mettack Attack | | | | | | Classification Accuracy at Perturbation Rate 15% | | Classification Accuracy at Perturbation Rate 20% | |
| Configuration | $\alpha$ | $\beta$ | $\gamma$ | $\eta$ | $\zeta$ | Graph Sparsity 22.7% Model Sparsity 67.6% | Graph Sparsity 60.4% Model Sparsity 98.2% | Graph Sparsity 22.6% Model Sparsity 67.5% | Graph Sparsity 60.3% Model Sparsity 98.2% |
| 1 | ✓ | ✓ | ✓ | ✓ | ✓ | **80.73** | **75.91** | **79.38** | **70.37** |
| 2 | ✓ | ✗ | ✓ | ✓ | ✓ | 80.23 | 73.69 | 78.72 | 69.97 |
| 3 | ✓ | ✓ | ✗ | ✓ | ✓ | 77.97 | 72.74 | 75.50 | 69.16 |
| 4 | ✓ | ✗ | ✗ | ✓ | ✓ | 78.42 | 72.08 | 74.09 | 68.86 |

Table 9: Comparison between the performance of ARGS and UGS with GCN, GIN, and GAT as backbone architectures on the clean Cora and Citeseer datasets.

| Dataset | Technique | Accuracy | Graph Sparsity | Model Sparsity |
|---|---|---|---|---|
| **GCN** | | | | |
| Cora | UGS | 80% | 26.52% | 74.20% |
| | ARGS | 80% | **40.17%** | **89.48%** |
| Citeseer | UGS | 70% | **64.56%** | **98.93%** |
| | ARGS | 70% | 64.27% | 98.88% |
| **GIN** | | | | |
| Cora | UGS | 79% | 26.82% | 73.93% |
| | ARGS | 79% | **33.73%** | **83.33%** |
| Citeseer | UGS | 68% | **66.05%** | **98.86%** |
| | ARGS | 68% | 64.27% | 98.86% |
| **GAT** | | | | |
| Cora | UGS | 80% | **65.35%** | **98.91%** |
| | ARGS | 80% | 64.32% | 98.93% |
| Citeseer | UGS | 70% | **67.61%** | **98.86%** |
| | ARGS | 70% | 64.24% | 98.86% |

## A.7 Performance Evaluation of ARGS on Clean Graphs

We finally evaluate the performance of ARGS on clean graphs. As shown in Table 9, ARGS can still find highly sparse GLTs in clean graphs. The lottery tickets found by ARGS achieve similar model and graph sparsity level when compared to UGS for the same classification accuracy on Cora and Citeseer datasets across three different GNN models. We assume the accuracy of UGS at 0% graph and 0% model sparsity as the baseline accuracy.

