# OpenReview forum: "Unveiling Adversarially Robust Graph Lottery Tickets"
_TMLR — Accepted by TMLR_

### Review · Reviewer_a5P8 · 2024-05-12

**Summary Of Contributions:**

This paper studies the adversarial robustness of graph lottery tickets. The authors investigate the resilience of GLTs against different structure perturbation attacks and observe that they are vulnerable and show a large drop in classification accuracy. Then they propose an adversarially robust graph sparsification (ARGS) framework that prunes the adjacency matrix and the GNN weights by optimizing a novel loss function capturing the graph homophily property and information associated with both the true labels of the train nodes and the pseudo labels of the test nodes. With extensive experiments, they verify that ARGS can find adversarially robust graph lottery tickets that are highly sparse yet achieve competitive performance under different training-time structure attacks.

**Audience:**

Yes

**Broader Impact Concerns:**

Could find any discussion of broader impacts.

**Claims And Evidence:**

No

**Requested Changes:**

1. What is the surrogate model used to generate adversarial perturbations?

2. What is the difference between the GLT setting and graph structure learning? For example [1].

3. The derived findings from the experiments are already found in the existing literature, including [2,3,4]. The discussion of related works should properly differentiate the similarities and dissimilarities with existing works.

4. What are the performances of the proposed method on clean graphs?

5. The experiments of
- Perturbation rates;
- Scale of datasets;
- GNN backbones;
- Up-to-date graph lottery ticket methods;

are all limited, making the generality of the conclusions suspected.

**References**

[1] Reliable Representations Make A Stronger Defender: Unsupervised Structure Refinement for Robust GNN, KDD'22.

[2] How does Heterophily Impact the Robustness of Graph Neural Networks? Theoretical Connections and Practical Implications. KDD'22.

[3] Understanding and Improving Graph Injection Attack by Promoting Unnoticeability, ICLR'22.

[4] Revisiting graph adversarial attack and defense from a data distribution perspective. ICLR'23.

**Strengths And Weaknesses:**

(+) The studied problem is new and seems to be interesting;

(-) Most of the findings seem to be well-studied in the literature;

(-) The presentation is confusing;

(-) The experiments may not be sufficient;

---

> ### Author Response · Authors · 2024-07-19
> **Author Response - 1**
>
> We will reference the reviewer with the identifier a5P8 as R3 and thank them for the helpful comments and insightful questions. We address the comments and requested changes by reviewer R3 pointwise. Comment n in the strength and weaknesses is denoted by R3Wn and the requested change n is denoted by R3Cn.
>
> Requested Changes:
> R3C1: We use DeepRobust, an adversarial attack library, to perform the PGD attack and MetaAttack and generate the perturbed graph adjacency matrix ${A}^{'}$. When performing these attacks we use surrogate models which are of the same type and architecture as that of the GNN model being attacked. For example, when attacking ARGS that is using a 2-layer GCN as the GNN model, the surrogate model is also a 2-layer GCN.
>
>
> R3C2: In the case of STABLE [1], the graph structure is refined using unsupervised learning, where the graph adjacency matrix is treated as learnable and pruned to potentially remove adversarial edges. Unlike STABLE, GLTs prune not only the graph adjacency matrix but also the GNN model itself. ARGS goes further by removing not only adversarial edges but also edges that have less impact on the performance of the GNN which are identified by the learned mask.
>
>
> R3C3: In our experiments, we have evaluated the adversarial robustness of UGS and ARGS both of which are techniques for identifying GLTs and to the best of our knowledge, no work in the existing literature has investigated the impact of adversarial attacks on GLTs.
>
>
> When analyzing the impact of adversarial attacks on the graph datasets, our findings are aligned with other works including those pointed out here and we have cited them accordingly. Some of these observations include (a) in homophilic graphs the attack tends to connect nodes with large attribute feature differences, and (b) the adversarial edges are not uniformly distributed throughout the graph adjacency matrix.
>
>
> R3C4: We evaluate the performance of ARGS on clean graphs and the results are reported in Table 9. We add the results here also for your perusal:
>
>
> | | | GCN | | | GIN | | | GAT | | |
> |----------|-----------|----------------|----------------------|----------------------|----------------|----------------------|----------------------|----------------|----------------------|----------------------|
> | Dataset  | Technique | Accuracy | Graph Sparsity | Model Sparsity | Accuracy | Graph Sparsity | Model Sparsity | Accuracy | Graph Sparsity | Model Sparsity |
> | Cora 	| UGS   	| 80%  | 26.52%           	| 74.20%            	| 79%     	| 26.82%           	| 73.93%           	| 80%      	| 65.35%           	| 98.91%           	|
> | 	| ARGS  	| 80% | 40.17%           	| 89.48%           	| 79%     	| 33.73%           	| 83.33%           	| 80%      	| 64.32%           	| 98.93%           	|
> | Citeseer | UGS   	| 70%  | 64.56%           	| 98.93%           	| 68%      	| 66.05%           	| 98.86%           	| 70%      	| 67.61%           	| 98.86%           	|
> | | ARGS  	| 70%  | 64.27%           	| 98.88%           	| 68%      	| 64.27%           	| 98.86%           	| 70%      	| 64.24%           	| 98.86%           	|
>
>
> Results show the lottery tickets found by ARGS achieve similar model and graph sparsity when compared to UGS for the same classification accuracy on Cora and Citeseer datasets across three different GNN models.
>
>
> R3C5: For our experiments, we have considered 5%, 10%, 15%, and 20% perturbation rates on 7 different datasets (both homophilic and heterophilic) including 2 large-scale datasets namely OGBN-ArXiv (170,000 nodes and 1.16 million edges), and OGBN-Products (2.5 million nodes and 61 million edges) for GCN, GAT, GIN, and DeeperGCN models. These results are presented in the main paper as well as the appendix. There are extensions [2] of UGS however none of these works addresses the problem of finding Graph Lottery Tickets (GLT) when the input graphs may have been adversarially attacked. In particular, the effort in [2] tries to improve the performance of UGS for clean graphs and is orthogonal to our work. Applying ARGS on this improved version of UGS will result in GLTs that are adversarially robust. We believe the current experiments (Fig. 5-14 and Tables 1-5, 7-9) show the effectiveness of our proposed technique.

---

> > ### Author Response · Authors · 2024-07-19
> > **Author Response 2**
> >
> > Weaknesses:
> >
> >
> > R3W2: To the best of our knowledge, this is the first work that analyzes the adversarial robustness of graph lottery tickets (GLTs). Our experimental results show that the existing GLT identification techniques like UGS are not able to combat adversarial attacks which are not previously been evaluated by other works.
> >
> >
> > R3W3: Please accept our apologies in case the presentation was confusing. We have updated the paper addressing the concerns from different reviewers. Now the flow of the paper is as follows: In the introduction we first introduce the problem statement and highlight the contributions of this work. In the related work section, we first introduce the GLT hypothesis and how UGS can identify GLTs. Then we introduce the different types of adversarial attacks on graph datasets and also the existing defenses. In the methodology section, we provide more technical details about GLTs and the attacks on GNNs and also preliminary knowledge which is used throughout the paper. In this section, we highlight the weakness of UGS, and the impact of adversarial attacks on graph datasets and finally introduce our proposed technique ARGS. In the evaluation section, we first explain the setup used. Next, we evaluate the adversarial robustness of ARGS and compare it with other baselines for homophilic graph datasets Cora, Citeseer, and PubMed. Next, we evaluate the robustness of ARGS for large-scale datasets OGBN-ArXiv, and OGBN-Products. We then perform adaptive attacks and evaluate the performance of ARGS. Following we also perform experiments for heterophilic datasets Chameleon and Squirrel. In addition to considering structural attacks, we also evaluate the performance of ARGS under structure + node feature attacks. Finally, we perform an ablation study and conclude the paper. Please let us know if we need to make further changes to this format.
> >
> > R3W4: In this work, we have performed experiments for both homophilic as well as heterophilic datasets (7 different datasets) for different GNN models. We have considered 4 different attacks as well as adaptive attacks. In terms of the scale of the datasets, as evident from Table 5, we have considered small as well as large datasets. In particular, we performed experiments for OGBN-ArXiv and OGBN-Products datasets which contain 169,343 nodes, 1,166,243 edges, and 2,449,029 nodes, 61,859,140 edges respectively.
> >
> >
> >
> >
> >
> >
> > [1]Reliable Representations Make A Stronger Defender: Unsupervised Structure Refinement for Robust GNN, KDD'22.
> >
> > [2] Rethinking Graph Lottery Tickets: Graph Sparsity Matters. ICLR 2023

---

> > > ### Comment · Reviewer_a5P8 · 2024-07-26
> > >
> > > Thank you for the rebuttal. It resolves most of my questions.

---

### Review · Reviewer_6wEC · 2024-05-26

**Summary Of Contributions:**

The authors present a technique to find Graph Lottery Tickets (GLT, pairs of sparsified input graphs and GNN models performing as well as the original input-model pair, yet with significantly reduced inference time) that are adversarially robust to structural poisoning attacks.
The proposed approach (ARGS) is a heuristic defense, based on a series of terms encouraging the removal of adversarial structural modifications from the graph.
Experiments showing that ARGS improves robustness to such attacks in practice on a series of benchmarks.

**Audience:**

Yes

**Broader Impact Concerns:**

None.

**Claims And Evidence:**

No

**Requested Changes:**

- More emphasis should be put on the fact that the proposed approach is designed for *poisoning* attacks, or experiments onto *evasion* attacks should be provided.
- More technical details about the baselines (UGS, defenses, attacks) should be provided.
- I missed how the perturbed A' (equation (7)) employed at training time is computed. These details should be clarified or given more prominence.
- What is the difference between UGS, ARGS, and the employed baseline for sparsity=0? Why do both UGS and ARGS outperform the baseline even when sparsity=0? Is this linked to the use of pseudo-labels?
- Adaptive and stronger attacks should be employed throughout the experimental evaluation.

**Strengths And Weaknesses:**

**Strengths.**

The paper appears to be the first to study the adversarial robustness of GLTs, finding that the standard sparsification algorithm (UGS) does not yield models that are inherently more robust to adversarial examples. The proposed approach (ARGS) appears to effectively present a way to find adversarially-robust GLTs (ARGLTs).

**Weaknesses.**

The paper is fairly hard to follow for readers not 100% familiar with the literature of adversarial robustness of GNNs. It would be fairly important to carefully introduce the technical details behind the baselines (UGS, first of all, but also the working mechanisms behind the attacks employed in the experimental evaluation).

On the methodological side, the design of ARGS appears to be mostly inspired by analysing the behaviour of some poisoning attacks on a couple of datasets (Figure 3), and then taking steps to heuristically counter their effect. How can the authors make sure that such techniques and observations generalize to further datasets and further attacks? The fact some algorithm components (equations (4) and (5)) change depending on the type of the dataset is not a good sign in this regard.

On a related note, the empirical evaluations seem to be mostly carried out on attacks and datasets similar to those employed for the analysis of Figure 3. On some plots, such as Figure 6, the baseline appears to be fairly weak, as it is outperformed also by UGS (not an adversarial defense) for many sparsity levels. Perhaps this is linked to the fact that the attacks used for these plots are not adaptive (section 4.4), and hence do not attack the proposed defense directly. It would be very important to see plots such as Figure 6 or 8 against stronger/adaptive attacks.

---

> ### Author Response · Authors · 2024-07-19
> **Author Response 1**
>
> We will reference the reviewer with the identifier 6wEC as R2 and thank them for the helpful comments and insightful questions. We address the comments and requested changes by reviewer R2 pointwise. Comment n in the weaknesses is denoted by R2Wn and the requested change n is denoted by R2Cn.
>
>
> Requested Changes:
> R1C1: As per suggestion, throughout the paper, we have mentioned that poisoning attacks are considered and the proposed technique ARGS is designed for combatting poisoning attacks. We have clarified the attack settings in the different sections and have highlighted them.
>
>
> R1C2: We have clarified this by adding new text in the paper. Specifically, for UGS, we introduce it in section 2.1 and then provide more technical details in section 3 (highlighted in blue). For the different types of attacks, we introduce them in section 2.2 and provide further details in section 3 and A.2. Also we have updated the details about the defenses in section 2.3.
>
>
> R1C3: We use DeepRobust, an adversarial attack library, to perform the PGD attack and MetaAttack and generate the perturbed graph adjacency matrix ${A}^{'}$. When performing these attacks we use surrogate models which are of the same type and architecture as that of the GNN model being attacked. For example, when attacking ARGS that is using a 2-layer GCN as the GNN model, the surrogate model is also a 2-layer GCN. We use Pytorch-Geometric~\citep{Fey/Lenssen/2019} to perform the PR-BCD and GR-BCD attacks on the OGBN-ArXiv and OGBN-Products datasets, respectively. This is also highlighted in the paper in section 4.1
>
>
> R1C4: Sparsity = 0 implies that no edges from the graph adjacency matrix and no weights from the GNN model are removed for ARGS and UGS. The difference between UGS and ARGS lies in the loss function being used for training the GNN. For the baseline defense techniques, we apply it to the perturbed graph dataset and obtain the classification accuracy which is used as the baseline accuracy for all the sparsity ratios starting from 0. Existing defense techniques do not iteratively prune the adjacency matrix or the GNN model weights. In some cases, ARGS outperforms the baseline defense technique’s classification accuracy at 0% sparsity. We believe this is potentially due to the use of pseudo labels.
>
>
> R1C5: As per suggestion, we have performed adaptive attacks for PubMed and OGBN-ArXiv dataset. These additional results are included in the paper in Table 3 and Table 4.

---

> > ### Author Response · Authors · 2024-07-19
> > **Author Response 2**
> >
> > Weaknesses:
> > R1W1: As per suggestion, we have added additional technical details about UGS, different attacks, and defenses to provide a more detailed context for the reader.
> >
> >
> > R1W2: Graph datasets can be broadly categorized into 2 types namely homophilic and heterophilic [1, 2, 3]. In this work, we consider these 2 broad categories of datasets, and the observations about the attacks we make are for graphs representing each of these categories. Therefore the observations made in this case do not represent a single dataset. As evident from the experiments, our proposed solution for the homophilic graph works across different small as well as large-scale datasets namely Cora, Citeseer, PubMed, OGBN-ArXiv, and OGBN-Products. Similar observations are also made for heterophilic graph datasets. A defender has knowledge about the dataset to defend being homophilic or heterophilic in nature. Note, in our proposed technique ARGS, only one component (whether to consider attribute features or positional features of the node) changes based on the nature of the dataset.
> >
> >
> > R1W3: Please note the baseline we previously used for Figure 6 is GARNET with 3-layer GCN as the GNN model. Our proposed technique ARGS uses 28-layer DeeperGCN as the GNN model. For a fair comparison we have conducted new experiments for GARNET with 28-layer DeeperGCN and as evident in the updated Figure 6, UGS does not outperform the baseline for sparsity levels greater than 5%, and its accuracy drastically reduces as we increase the sparsity levels. Also, other Figures in the paper show that UGS performs much worse than the baseline. As per your suggestion, we also performed adaptive attacks on OGBN-ArXiv and PubMed datasets, and the results are included in Tables 3 and 4.
> >
> >
> > [1] Zhu, Jiong, et al. "How does heterophily impact the robustness of graph neural networks? theoretical connections and practical implications." SIGKDD 2022.
> > [2] Deng, Chenhui, et al. "Garnet: Reduced-rank topology learning for robust and scalable graph neural networks." Learning on Graphs Conference. PMLR, 2022.
> > [3] McCallum, Andrew Kachites, et al. “Automating the construction of internet portals with machine learning.” Information Retrieval, 2000

---

> > > ### Comment · Reviewer_6wEC · 2024-07-31
> > > **Thank you for your response**
> > >
> > > I thank the authors for their reply. The quality of the presentation has significantly improved. I think the paper successfully shows that ARGS finds GLTs that are more robust than those found by UGS. However, its relative effectiveness with respect to other defenses for poisoning attacks on GNNs is still unclear to me. Figure 6 still shows that even UGS performs better than the chosen baseline defense, and the authors agreed this could be due to pseudo-labels. To me, this indicates that the comparison may be unfair. Ideally, more defenses described in the new section 2.3 should be included in the experimental comparisons to get a full picture.

---

### Review · Reviewer_i6qx · 2024-07-05

**Summary Of Contributions:**

This paper investigates the resilience of Graph Lottery Tickets (GLTs) against adversarial attacks and proposes the Adversarially Robust Graph Sparsification (ARGS) framework to enhance their robustness. The paper first evaluates the vulnerability of GLTs identified by Unified Graph Sparsification (UGS) against various adversarial attacks. The results demonstrate that GLTs show a significant drop in classification accuracy under such attacks. The authors propose the ARGS framework, which prunes the graph adjacency matrix and GNN weights by optimizing a novel loss function that captures the graph homophily property and information from both true and pseudo labels. This framework iteratively refines the graph and model to achieve adversarial robustness. The paper compares the performance of ARGS with other state-of-the-art adversarial defense techniques like GNNGuard, GARNET, and ProGNN. The results indicate that ARGS achieves better accuracy and efficiency in terms of inference MACs, demonstrating its effectiveness in identifying highly sparse yet robust GLTs.

**Audience:**

Yes

**Broader Impact Concerns:**

No broader impact concern.

**Claims And Evidence:**

Yes

**Requested Changes:**

1. Provide efficiency analysis of the pruning method and compare with the baselines.

**Strengths And Weaknesses:**

Strengths:
1. The paper provides a thorough analysis of the vulnerabilities of GLTs under various adversarial attacks, highlighting the need for improved robustness in graph neural networks.
2. The proposed ARGS framework is a novel approach that effectively combines graph and model pruning to enhance adversarial robustness, making it a significant contribution to the field.

Weakness:
1. The ARGS framework involves a complex optimization process, which may impedes its application.

---

> ### Author Response · Authors · 2024-07-19
> **Author Response**
>
> We will reference the reviewer with the identifier i6qx as R1 and thank them for the helpful comments and insightful questions. We address the comments and requested changes by reviewer R1 pointwise. Comment n in the weaknesses is denoted by R1Wn and the requested change n is denoted by R1Cn.
>
>
> R1W1:
> The proposed method uses three hyperparameters, namely, β λ1, and λ2. The value of α and γ in equation 7 is always 1 to enable uniform removal of edges surrounding the training as well as the test nodes. We have included these variables in the equation since they are set to 0 in the ablation study, to show the impact of different terms in the loss function on the performance of ARGS. We have evaluated ARGS across 7 different datasets including OGBN-Products which contains 2,449,029 nodes and 61,859,140 edges. Experimental results show the effectiveness of the proposed technique.
>
>
> R1C1:
> We provide the efficiency analysis of our pruning method and its comparison with baseline techniques in terms of the number of MACs during inference. Results are included in the paper in section 4 (Fig.5, 6, 8, and Table 1 and 2). We also summarize the results of ARGS and STRG for Cora, Citeseer, and OGBN_ArXiv datasets here:
>
>
>
>
> | Dataset (GNN backbone, Attack, Perturbation rate) | Defense Technique | Inference MAC Count (lower better) |
> |---------------------------------------------------|-------------------|---------------------|
> | Cora (GCN, PGD, 20%)                              | ARGS              | **78.78M**              |
> |                                                   | STRG              | 1832.14M            |
> | Cora (GIN, PGD, 20%)                              | ARGS              | **246.52M**             |
> |                                                   | STRG              | 1841.05M            |
> | Cora (GAT, PGD, 20%)                              | ARGS              | **316.97M**             |
> |                                                   | STRG              | 14765.65M           |
> | Citeseer (GCN, Mettack, 20%)                      | ARGS              | **54.69M**              |
> |                                                   | STRG              | 4006.91M            |
> | Citeseer (GIN, Mettack, 20%)                      | ARGS              | **45.95M**              |
> |                                                   | STRG              | 4013.39M            |
> | Citeseer (GAT, Mettack, 20%)                      | ARGS              | **363.24M**             |
> |                                                   | STRG              | 32144.71M           |
> | OGBN_ArXiv(DeeperGCN, PRBCD, 10%)                 | ARGS              | **4606.56M**             |
> |                                                   | GARNET            | 83755.69M            |

---

### Decision · Action_Editor_tjy6 · 2024-08-11

**Recommendation:** Accept with minor revision

**Comment:**

My understanding of the key contribution of this work is that it demonstrates the existence of adversarially robust graph lottery tickets and proposes a technique to identify such tickets. There is a consensus among the reviewers that this could be an interesting contribution. However, since the study proves this existence "empirically," the paper's evaluation needs to be comprehensive and extensive.

In this regard, one reviewer has expressed concern about the lack of evaluation of the proposed method against strong defense baselines. While I don't believe this work needs to be evaluated against the "strongest possible defenses," the reviewer's concern seems to be more about the evaluation configuration, which may have led to an unfair comparison.

I believe the reviewer's point is fair, and I believe it should be addressed during the minor revision process. Other than that, I could not foresee any other problems.

**Audience:**

The paper presents adversarially robust graph lottery tickets, a concept that has been studied in feedforward networks but is less explored in graph neural networks. This work could serve as a reference point for those interested in transferring knowledge graph models.

**Claims And Evidence:**

Through the rounds of author-reviewer discussion, the manuscript has been significantly improved, particularly in terms of related works and clarity. The manuscript now provides a review of prior work on defenses, elaborates on the concept of graph lottery tickets, and details the experimental setup. Additionally, the evaluation section has been expanded to include more attacks and defenses.